# Neural circuit mechanisms for steering control in walking *Drosophila*

Aleksandr Rayshubskiy[1†], Stephen L Holtz[1], Alexander S Bates[1],
Quinn X Vanderbeck[1], Laia Serratosa Capdevila[2], Victoria Rockwell[1],
Rachel Wilson[1]*

[1]Department of Neurobiology, Harvard Medical School, Boston, United States;
[2]Aelysia LTD, Bristol, United Kingdom

## eLife Assessment

This **important** work investigates how orientation signals detected in higher brain areas may be transformed into motor responses in behaving animals. The authors characterize two types of descending neurons (DNs) that connect the brain to motor units and are involved in different aspects of turning control. They further show that orientation signals act by preferentially increasing relative stimulation onto left- or right-turn-inducing DNs. These **compelling** results, together with the independent work that they have inspired, represent significant progress in our understanding of mechanisms of animal navigation.

**\*For correspondence:**
rachel_wilson@hms.harvard.edu

**Present address:** †Rowland Institute at Harvard University, Cambridge, United States

**Abstract** Orienting behaviors provide a continuous stream of information about an organism's sensory experiences and plans. Thus, to study the links between sensation and action, it is useful to identify the neurons in the brain that control orienting behaviors. Here, we describe descending neurons in the *Drosophila* brain that predict and influence orientation (heading) during walking. We show that these cells have specialized functions: whereas one cell type predicts sustained low-gain steering, the other predicts transient high-gain steering. These latter cells integrate internally directed steering signals from the head direction system with stimulus-directed steering signals from multimodal sensory pathways. The inputs to these cells are organized to produce 'see-saw' steering commands, so that increasing output from one brain hemisphere is accompanied by decreasing output from the other hemisphere. Together, our results show that internal and external drives are integrated to produce descending motor commands with different timescales, for flexible and precise control of an organism's orientation in space.

## Introduction

Orienting behaviors provide basic clues about an organism's perceptions and memories (*Schöne, 2014*). For example, animals will often orient readily toward novel or attractive stimuli, so orientation can provide information about perceived salience or valence (*Simion and Shimojo, 2007*; *Sokolov, 1960*). Animals also actively stabilize their orientation relative to the environment, and the dynamics of this behavior provide clues about mechanisms of sensorimotor integration (*Götz, 1972*). Animals can estimate their orientation over time by keeping track of their rotational movements, providing a way to study mechanisms of working memory (*Goodridge et al., 1998*; *Seelig and Jayaraman, 2015*). Finally, animals can orient toward a remembered target in the environment, and so orientation can provide insights into spatial learning (*Summerfield et al., 2006*).

Ideally, we would like to study these behaviors by identifying the neurons in the brain that generate orienting movements. In principle, this would allow us to work backward from these neurons to

understand the convergence of upstream orienting cues from sensory regions, as well as regions involved in higher cognition. Recent work has identified descending neurons in the mammalian brainstem that influence orienting during locomotion ('steering'): these cells are active when the animal is executing a locomotor turn, and their unilateral activation can drive turning during locomotion (*Cregg et al., 2020*; *Usseglio et al., 2020*). These brainstem descending neurons respond to perturbations of striatal neurons (*Cregg et al., 2024*), but we still do not fully understand how brain dynamics generate locomotor dynamics, in part because of the anatomical complexity of the mammalian brain, and the challenges involved in obtaining neurophysiological data with high temporal and spatial resolution.

Some of these experimental challenges are easier to overcome in *Drosophila melanogaster*. In particular, the advent of large-scale *Drosophila* connectomes (*Cheong et al., 2023*; *Dorkenwald et al., 2023*; *Dorkenwald et al., 2022*; *Eckstein et al., 2024*; *Li et al., 2020*; *Phelps et al., 2021*; *Scheffer et al., 2020*; *Schlegel et al., 2023*; *Zheng et al., 2018*) has made it easier to identify descending neurons (DNs) that might be involved in specific behaviors (*Simpson, 2024*). Moreover, it is possible to perform targeted in vivo electrophysiological recordings from specific cells in these connectomes, and to relate these recordings to ongoing locomotor behaviors or brain dynamics. Several recent studies have linked specific *Drosophila* DNs to escape (*Lima and Miesenböck, 2005*; *von Reyn et al., 2014*), backward walking (*Bidaye et al., 2014*), grooming (*Guo et al., 2022*; *Hampel et al., 2015*), flight (*Suver et al., 2016*), and landing (*Ache et al., 2019*). A recent imaging study also found that multiple DNs are active when a walking fly executes voluntary turns, but technical limitations prevented most of these DNs from being matched with cells in the connectome (*Aymanns et al., 2022*). We, therefore, set out to identify DNs in the connectome with clear functional roles in steering.

Here, we describe two DN types in the *Drosophila* brain that predict steering during walking. Using targeted in vivo single-cell electrophysiological recordings in walking flies, we find that one type (DNa02) predicts high-gain steering, while the other (DNa01) predicts low-gain steering. DNa02 is recruited first, on average, and its activity is more transient. DNa02 lies two synapses downstream from head direction neurons, and we demonstrate that these DNs are recruited during memory-directed steering that is guided by the head direction system. DNa02 also receives multimodal sensory input from reinforcement learning centers, and we show that this DN is recruited by multimodal sensory inputs with differing valence. Using dual recordings from both copies of DNa02, we show that the fly's turning velocity is linearly related to the right-left difference in DNa02 activity; then, through analysis of large-scale connectome data, we show how the feedforward inputs to this cell from the head direction system and from sensory pathways suggest a 'see-saw' model of steering control, where excitation of one DN copy is accompanied by inhibition of the contralateral copy. Together, our results show how a motor control problem (steering) can be re-framed as a problem of generating a specific pattern of brain activity via converging sensory and cognitive pathways.

## Results

### Two descending neuron types that predict different features of steering

DNa01 and DNa02 project to all three leg neuromeres in the ventral nerve cord (*Figure 1A*), where they form selective connections onto leg-control networks (*Braun et al., 2024*; *Cheong et al., 2023*). These DNs are anatomically similar to cricket neurons that are active during turning (*Brodfuehrer and Hoy, 1990*; *Staudacher and Schildberger, 1998*; *Zorović and Hedwig, 2011*), and a calcium imaging study reported that DNa01 is asymmetrically active in the ventral nerve cord when flies are turning (*Chen et al., 2018*). On the other hand, an optogenetic perturbation study reported that bilateral activation of DNa01 or DNa02 drives a global increase in walking (*Cande et al., 2018*). Together, these observations suggested a role in locomotor control, but the specific functions of these cells were unclear. In analyzing the ventral nerve cord connectome (*Cheong et al., 2023*), we noticed that DNa02 makes more direct synaptic connections onto motor neurons, as compared to DNa01 (*Figure 1B*). Moreover, very few of the postsynaptic cells targeted by these two DN types are shared in common (*Cheong et al., 2023*; *Figure 1C*). These connectome analyses suggested that, if both DNs are involved in locomotor control, their functions are likely distinct.

To investigate this idea, we made genetically targeted whole-cell recordings from both DN types in flies walking on a spherical treadmill. We made single-cell recordings as well as dual recordings

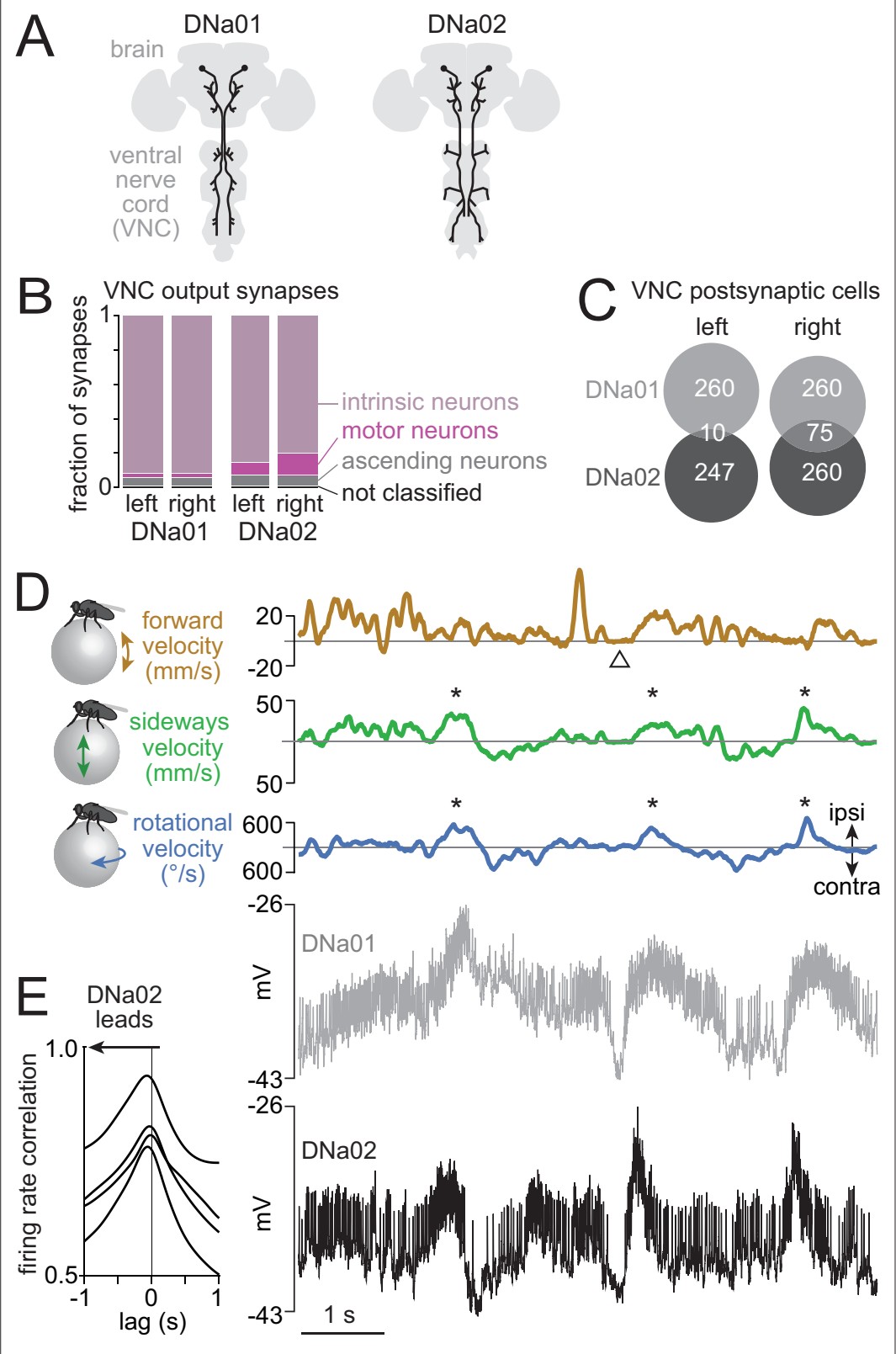

**Figure 1.** DNa01 and DNa02 predict different features of steering. (**A**) Schematic morphologies of these descending neurons (DNs) (*Namiki et al., 2018*), which have distinct projections in the ventral nerve cord (VNC). (**B**) Output synapses in the VNC (*Cheong et al., 2023*; *Marin et al., 2024*; *Takemura et al., 2023*), categorized by postsynaptic cell type. (**C**) Number of VNC cells postsynaptic to each DN. Both DN types are restricted to one side

*Figure 1 continued on next page*

*Figure 1 continued*

of the VNC. (**D**) Example recording of the fly's movement on a spherical treadmill, together with a simultaneous dual recording from DNa01 and DNa02, both on the left brain hemisphere. Both cells are depolarized just before the fly steers in the ipsilateral direction (asterisks). Both cells are hyperpolarized when the fly briefly stops moving (arrowhead). Here and elsewhere, ipsi and contra are defined relative to the recorded DN(s); thus, in this case, ipsilateral is left. (**E**) Correlation between DNa01 and DNa02 firing rates, as a function of the time lag between the two cells. Each line is a different paired recording (four flies).

The online version of this article includes the following figure supplement(s) for figure 1:

**Figure supplement 1.** Discriminating DNa01 from DNa02 in dual recordings.

**Figure supplement 2.** Forward and reverse linear filters for DNa01 and DNa02 neurons.

from these DNs to understand their correlation with behavior, and with each other (*Figure 1D*). To perform dual recordings, we used a combination of transgenes that targets both cells (*Figure 1— figure supplement 1*). In single-cell recordings, we defined the ipsilateral side of the body as the side of the recorded DN, allowing us to pool data from cells recorded on the right and left sides of the brain. During each recording, we tracked the fly's fictive locomotion in all three axes of motion (forward, sideways, and rotation). When a walking fly steers, it tends to rotate while stepping sideways (*Figure 1D*, *Figure 1—figure supplement 2*; *DeAngelis et al., 2019*; *Katsov et al., 2017*). We found that both DN types tended to increase their firing rate just before a steering maneuver. Notably, DNa02 had more transient fluctuations, as compared to DNa01 (*Figure 1D*, *Figure 1—figure supplement 2*). Moreover, spike rate fluctuations in DNa02 typically preceded the spike rate fluctuations in DNa01 (*Figure 1D and E*).

For both DN types, we computed the linear filters that describe the relationship between firing rate and behavioral dynamics. These filters tell us how behavior changes after a brief increase in firing rate – i.e., a firing rate 'impulse.' To compute these filters, we cross-correlated the firing rate with the fly's rotational velocity, sideways velocity, or forward velocity. We then normalized each filter by the autocorrelation in the cell's firing rate (*Figure 1—figure supplement 2*). These filters showed that DNa01 and DNa02 firing rate increases were typically followed by relatively large changes in rotational and sideways velocity (*Figure 2A and B*). DNa01 and DNa02 firing rate increases were not consistently followed by large changes in forward velocity (*Figure 2C*).

We also found several interesting functional differences between DNa01 and DNa02. First, steering filters (i.e. rotational and sideways velocity filters) were larger for DNa02 (*Figure 2A and B*). This means that an impulse change in firing rate predicts a larger change in steering for this neuron. In other words, this result suggests that DNa02 operates with higher gain. This may be related to the fact that DNa02 makes more direct output synapses onto motor neurons (*Figure 1B*). Moreover, we found that DNa02 steering filters were biphasic, whereas DNa01 steering filters were monophasic (*Figure 2A and B*). A biphasic filter converts a sustained input into a transient output, while a monophasic filter converts a sustained input into a sustained output. This result suggests that DNa02 produces more transient changes in steering, as compared to DNa01. This may be related to the fact that DNa02 and DNa01 target largely non-overlapping populations of postsynaptic cells in the ventral nerve cord (*Figure 1C*). Finally, we found that steering filters for DNa02 were more accurately predictive of behavior – i.e., when we convolved each filter with the cell's spike train, the DNa02 filters produced a better prediction of the fly's subsequent steering, as compared to the DNa01 filters (*Figure 2D and E*). This might also be attributable to the fact that DNa02 makes more direct output synapses onto motor neurons (*Figure 1B*).

To recap, DNa02 fires more transiently than DNa01, and it is also recruited before DNa01 (*Figure 1D and E*). But even if we normalize for the firing rate dynamics of each cell type, the behavioral impulse response is also more transient for DNa02 than for DNa01 (*Figure 2A and B*). In other words, DNa02 impulses arrive in relatively transient packets *and* each DNa02 impulse predicts a relatively transient change in behavior; for both these reasons, DNa02 is associated with more transient steering events, as compared to DNa01.

Next, we looked more closely at the magnitude of the behavioral responses associated with each cell type. Specifically, we compared neural activity with behavior 150 ms later, to account for the average lag between neural activity and behavior (*Figure 2A–C*); this lag may reflect a biological delay

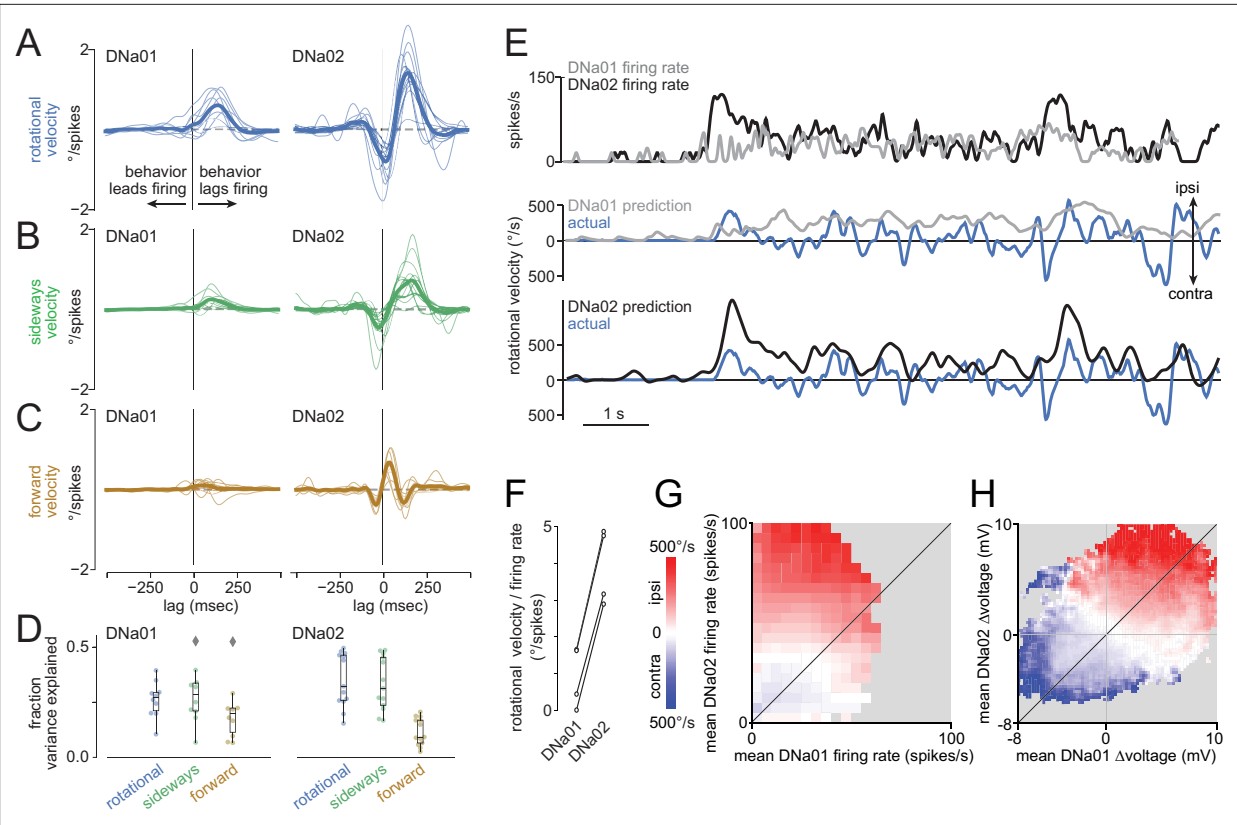

**Figure 2.** DNa02 and DNa01 have different firing rate dynamics and steering gain. (**A**) Linear filters describing the relationship between rotational velocity and descending neuron (DN) firing rate. Specifically, these filters describe the average rotational velocity impulse response, given a delta function (unit impulse) in firing rate. Thin lines are filters for individual flies (n=13 single-cell recordings for DNa02, n=10 single-cell recordings for DNa01). Thick lines are averages. (**B,C**) Same for sideways and forward velocity. (**D**) Variance explained by each filter type. Data points are flies. Boxplots show interquartile range (IQR), whiskers show range of data (except for diamonds, observations >1.5 × IQR). See also *Figure 1—figure supplement 1*. (**E**) The rotational velocity filter for each cell was convolved with its firing rate to generate rotational velocity predictions. Whereas DNa02 over-predicts large rapid steering events, DNa01 under-predicts these events. (**F**) Slope $m$ of the linear regression fitting rotational velocity $v_r$ to each cell's firing rate $f$ ($v_r = mf + b$). The difference between cell types is significant (p=$10^{-4}$, paired t-test, n=4 flies). (**G**) Binned and averaged rotational velocity as a function of DNa01 and DNa02 firing rate, for an example paired recording. Rows represent the relationship between DNa01 and turning, for each level of DNa02 activity. Columns represent the relationship between DNa02 and turning, for each level of DNa01 activity. Gray bins have too few datapoints to plot. (**H**) Same as (**G**) but for voltage changes rather than firing rate changes.

and/or the inertia of the spherical treadmill. We observed that rotational velocity was a relatively steep function of DNa02 activity, but a comparatively shallower function of DNa01 activity (*Figure 2F–H*). This analysis supports the conclusion that there is a higher-gain relationship between DNa02 activity and steering.

## Right-left differences in descending neuron activity predict rotational velocity

Next, to determine how steering depends on DNa02 neurons on both sides of the brain, we made dual recordings from DNa02 on the right and left (*Figure 3A*). We found that rotational velocity was consistently related to the difference in right-left firing rates (*Figure 3B*). This relationship was essentially linear through its entire dynamic range, and was consistent across paired recordings (*Figure 3C*). It was also consistent during backward walking, as well as forward walking (*Figure 3—figure supplement 1*). We obtained similar results in dual recordings from DNa01 neurons (*Figure 3—figure supplement 2*).

We also extended our linear filter analysis to our dual DNa02 recordings. Specifically, we computed each cell's linear filter, and we then convolved each filter with the cell's spike train to predict the fly's

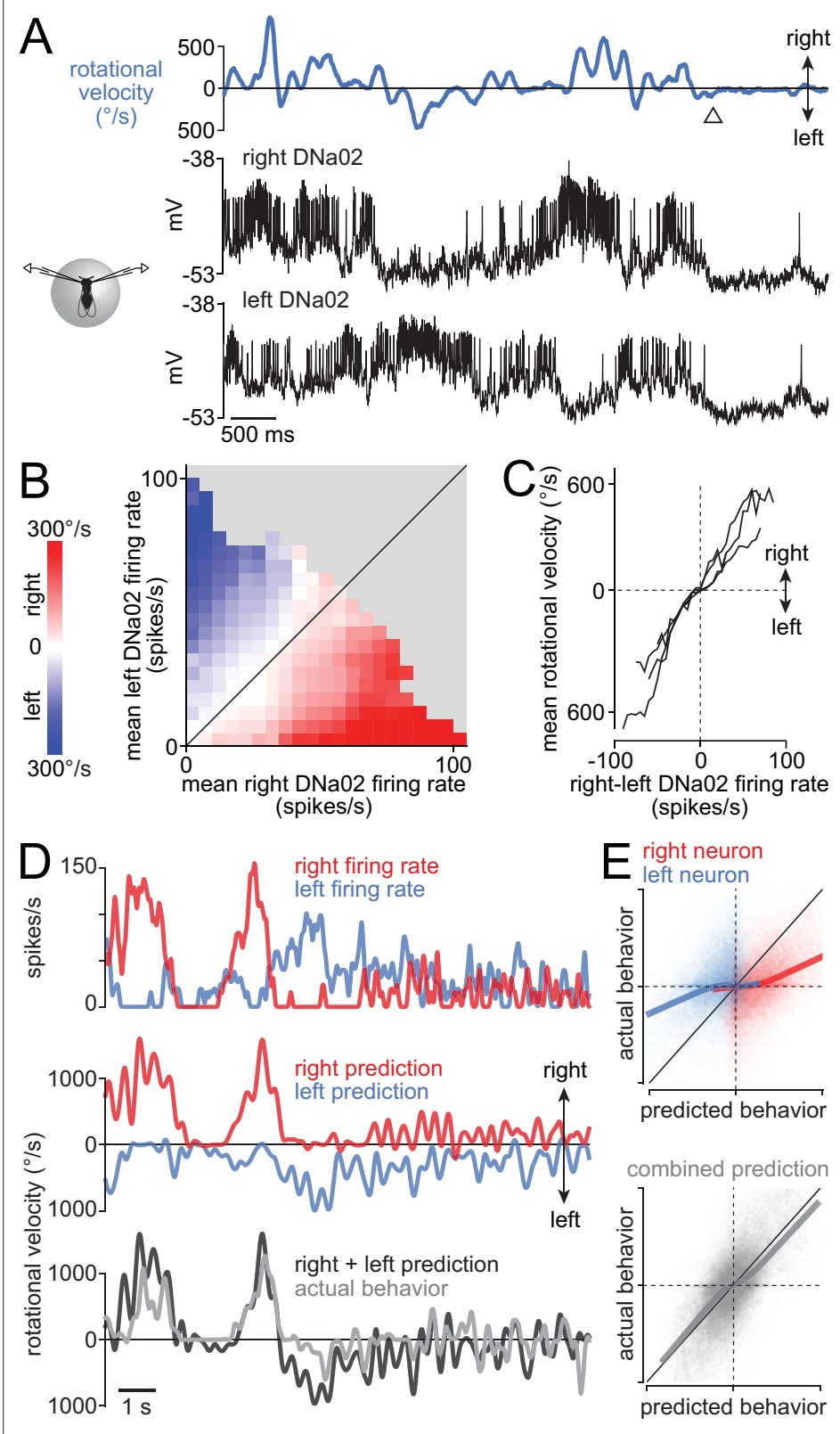

**Figure 3.** Bilateral differences in descending neuron (DN) firing rate predict steering. (**A**) Example bilateral recording from DNa02. The neurons are anti-correlated when the fly is turning and hyperpolarized when the fly stops moving (arrowhead). (**B**) Binned and averaged rotational velocity for each value of bilateral DNa02 firing rates for an example paired recording. (**C**) Mean rotational velocity for each value of the bilateral firing rate difference.

*Figure 3 continued on next page*

*Figure 3 continued*

Each line is a different fly (n=4 flies). (**D**) The rotational velocity filter for each cell was convolved with its firing rate to predict rotational velocity. Combining the predictions of the two cells (with equal weighting) generates a good prediction. (**E**) Top: predicted rotational velocity versus actual rotational velocity, for both single-cell predictions for an example experiment. Bottom: the dual-cell prediction for this experiment. Thick lines are LOWESS fits. Each axis ranges from 800 °/s rightward to 800 °/s leftward, and dashed lines denote 0 °/s.

The online version of this article includes the following figure supplement(s) for figure 3:

**Figure supplement 1.** Forward versus backward walking.

**Figure supplement 2.** Dual recordings from the right and left copies of DNa01.

rotational velocity. By adding the predictions from simultaneously recorded right-left pairs, we found that we could accurately predict turns in both directions (*Figure 3D and E*).

To summarize, these dual recordings demonstrate that steering is consistently predicted by right-left differences in DNa02 firing rate. This means that, in principle, turning could be driven by either ipsilateral excitation or contralateral inhibition. As we will see, the neural networks in the brain presynaptic to these DNs are organized so as to recruit both ipsilateral excitation and contralateral inhibition.

## Steering toward internal goals

Having identified steering-related DNs, we proceeded to investigate the brain circuits that provide input to these DNs. Here, we decided to focus on DNa02, as this cell's activity is predictive of larger steering maneuvers. First, we asked whether DNa02 responds to changes in perceived head direction relative to an internal goal. Head direction neurons (EPG neurons) (*Seelig and Jayaraman, 2015*) can influence steering (*Giraldo et al., 2018*; *Green et al., 2019*) via central complex neurons that compare head direction signals with internal goal signals (*Beetz et al., 2022*; *Martin et al., 2015*; *Mussells Pires et al., 2024*; *Shiozaki et al., 2020*; *Westeinde et al., 2024*). The central complex itself contains no DNs (*Hulse et al., 2021*; *Namiki et al., 2018*), but there are strong anatomical pathways from the central complex to DNa02 (*Hulse et al., 2021*; *Mussells Pires et al., 2024*; *Rayshubskiy et al., 2020*; *Westeinde et al., 2024*). This motivated our hypothesis that DNa02 participates in steering toward internal goals.

To investigate the functional coupling between the head direction system and steering system, we made whole-cell recordings from DNa02 while monitoring and manipulating EPG neurons. To track the dynamics of the head direction system in real time, we imaged GCaMP6f in the entire EPG ensemble. Meanwhile, we micro-stimulated central complex neurons called PEN1 neurons (*Green et al., 2019*) by expressing the ionotropic ATP receptor P2X$_2$ in PEN1 neurons under Gal4/UAS control (*Lima and Miesenböck, 2005*) and puffing ATP onto their dendrites (*Figure 4A*). PEN1 neurons relay information about the fly's angular velocity to EPG neurons, and so micro-stimulating PEN1 neurons causes the head direction system to register a fictive behavioral turn.

As expected from previous work, we found that there is a single bump of activity in the EPG ensemble whose angular velocity generally correlates with the fly's angular velocity (*Green et al., 2017*; *Seelig and Jayaraman, 2015*; *Turner-Evans et al., 2017*). The position of the EPG bump constitutes a working memory of the fly's heading direction. Again, consistent with previous work, we found that briefly micro-stimulating PEN1 neurons causes this bump to 'jump' to a new location (*Green et al., 2019*). As a negative control, we confirmed that bump jumps rarely follow ATP injection when the Gal4 transgene is omitted (*Figure 4B*). We only analyzed trials where the EPG bump was relatively stable prior to ATP injection, implying that the fly thinks it is walking in a straight line and is trying to steer toward an internal goal (*Green et al., 2019*). During epochs of goal-directed steering, the EPG bump jump should create a mismatch (error) between the fly's internal goal and its perceived head direction. Accordingly, we find that the fly typically executes a compensatory behavioral turn shortly after the EPG bump jump, as reported previously (*Green et al., 2019*). This compensatory turn brought the EPG bump back to its initial location in 40% of trials, thereby re-orienting the fly toward its inferred internal goal. Notably, in trials where the ATP injection caused the bump to jump clockwise, the behavioral turn was typically rightward, which causes the bump to flow counter-clockwise (*Figure 4C and D*). Conversely, on trials where the injection caused the bump to jump counter-clockwise, the average behavioral turn was leftward, which causes the bump to flow clockwise

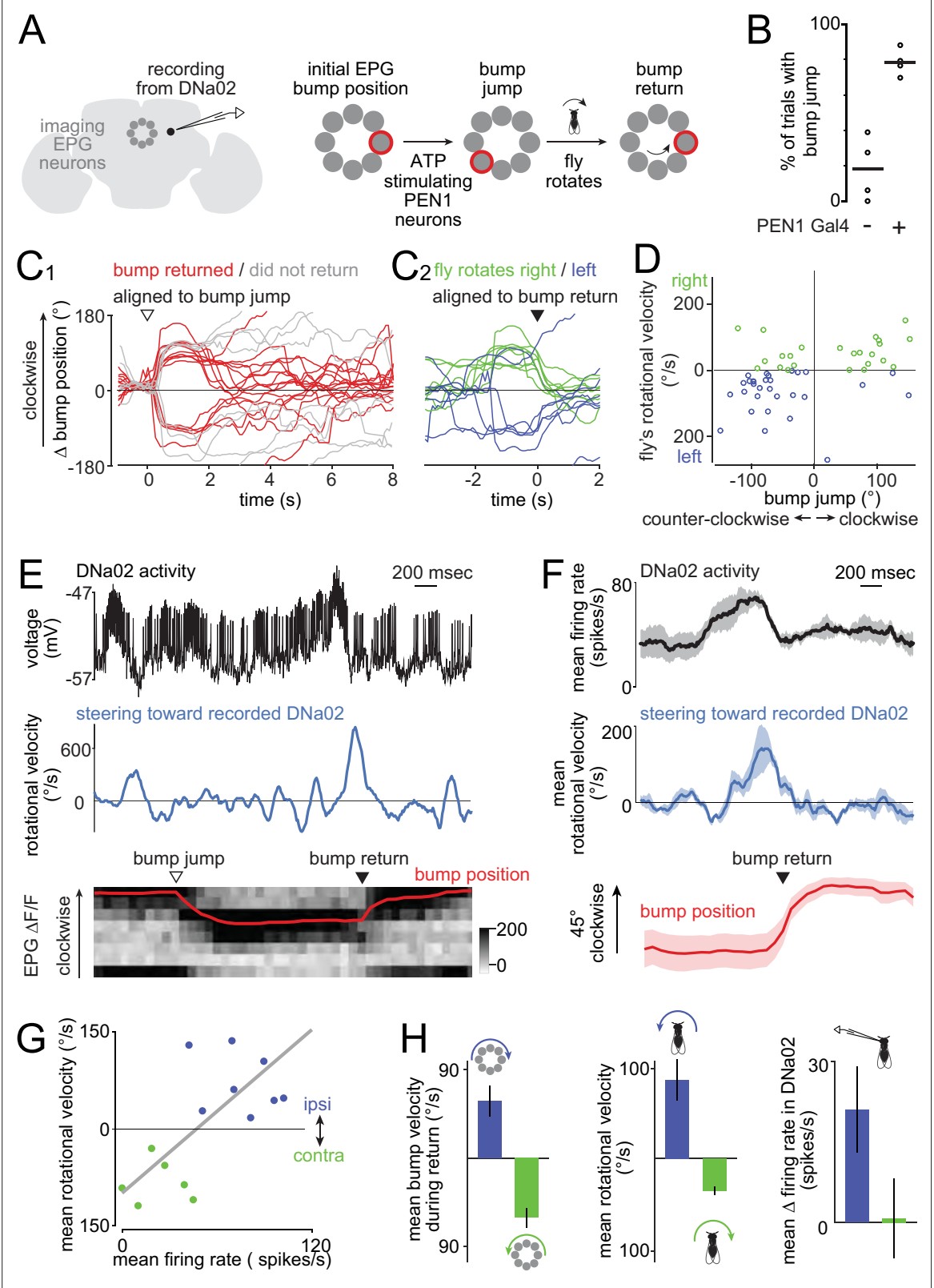

**Figure 4.** DNa02 participates in compass-directed steering. (**A**) A bump of activity rotates around the ensemble of EPG neurons (compass neurons) as the fly turns. We microstimulate PEN1 neurons with ATP to evoke a bump jump. This is followed by a compensatory behavioral turn that returns the bump to its initial location. Here, the brain is depicted as viewed from the back (as during our experiments). In this view, a clockwise fly rotation produces a counter-clockwise bump rotation. (**B**) Percentage of trials with a bump jump after the ATP puff. Each point is a fly (each genotype n=4).

*Figure 4 continued on next page*

*Figure 4 continued*

In controls lacking PEN1 Gal4, ATP only occasionally preceded bump jumps, which is likely coincidental, as the bump often moves spontaneously. (**C1**) Change in bump position after ATP. The bump returns to its initial position in many trials (red) but not all (gray) in this sample experiment. (**C2**) Red trials from C1 aligned to the time of maximal bump return speed, color-coded by the fly's steering direction during bump return. (**D**) The fly's rotational velocity during the bump return, plotted against the initial bump jump. Clockwise bump jumps are generally followed by rightward turns (which drive bump return via a counterclockwise path), and vice versa. Clockwise bump jumps (18 trials) and counter-clockwise bump jumps (35 trials) are followed by significantly different mean rotational velocities (p<0.05, t-tailed t-test, pooling data from 4 flies). (**E**) Example trial. Top: DNa02 activity. Middle: fly's rotational velocity toward the left side, i.e., the side of the recorded DNa02 neuron. Bottom: grayscale EPG ΔF/F over time, where each row is a 45°-sector of the compass, and red is bump position. ATP causes a bump jump (arrowhead). Then, the left copy of DNa02 bursts, the fly turns left, and the bump returns via a clockwise path (arrowhead). (**F**) Trials where the bump returned to its initial location via a clockwise path were aligned to the time of peak bump speed. Trials were averaged within a fly, and then across flies (mean ± SEM across flies, n=4 flies). (**G**) In an example experiment, rotational velocity correlates with DNa02 firing rate on a trial-to-trial basis ($R^2$=0.51, p=3 × 10$^{-3}$, two-tailed t-test). Trials are color-coded as in (**D**). In all other experiments, p was also <0.05. (**H**) Data sorted by the direction of the bump's return (mean ± SEM across flies, n=4 flies). Whereas clockwise (blue) bump returns were typically preceded by leftward turning, counter-clockwise (green) bump returns were preceded by rightward turning, as expected. On average, the left copy of DNa02 was only excited on trials where the bump moved clockwise, meaning the fly was turning left.

(*Figure 4C and D*). Thus, the compensatory turn was in the correct direction to return the bump to its original location via the shortest path. This implies that the mechanism that triggers the compensatory turn is sensitive to the direction of the error (positive or negative). These results are consistent with previous work (*Green et al., 2019*) except that we observed a more variable delay between the bump jump and the compensatory turn.

Throughout each experiment, we performed a whole-cell recording from DNa02. Notably, we found that DNa02 cells often fired a burst of spikes just before the fly performed its compensatory behavioral turn to bring the EPG bump back to its initial location (*Figure 4E*). To quantify this effect, we identified the time point in each trial where the bump's return speed was maximal. We used this time point to align the data across trials to account for the fact that the bump's return had different latencies on different trials (*Figure 4C and E*). Because we always recorded from DNa02 cells on the left, we focused on the trials where we expected a leftward behavioral turn – i.e., trials where the bump returned via a clockwise path. In these trials, we found that the bump jump was typically followed by increased DNa02 firing on the left, and then a leftward steering maneuver, and then a clockwise return of the bump (*Figure 4F*). Moreover, we found that DNa02 could predict much of the trial-to-trial variability in the magnitude of compensatory behavioral turns (*Figure 4G*).

Trials where the bump returned in the opposite direction (counter-clockwise) provide a negative control. In these trials, the fly's behavioral turn was typically rightward rather than leftward (*Figure 4E and H*). Accordingly, we found no DNa02 firing rate increase on the left side (*Figure 4H*). This shows that DNa02 is not activated bilaterally and thus nonspecifically. Rather, DNa02 is recruited specifically on the side that predicts the direction of the behavioral turn.

In summary, our results show that DNa02 activity can predict the magnitude and direction of steering maneuvers guided by the head direction system. This finding argues that the pathways from head direction system to DNa02 are functionally strong, and they are recruited at the appropriate time to mediate steering commands issuing from the central complex.

## Stimulus-directed steering

Next, we asked whether DNa02 is functionally engaged in stimulus-directed steering, which we define as steering directly toward (or directly away from) a sensory stimulus. To generate an attractive sensory stimulus, we expressed the channelrhodopsin variant CsChrimson in olfactory receptor neurons under the control of an *Orco-LexA* transgene. We then stimulated the right or left antennae independently using two thin optical fibers. This fictive odor stimulus produced steering toward the stimulated antenna (*Figure 5A*), consistent with previous studies showing that flies often turn toward a lateralized odor (*Borst and Heisenberg, 1982*; *Gaudry et al., 2013*). We found that these lateralized stimuli also produced asymmetric responses in DNa02, with higher firing rates on the ipsilateral side (*Figure 5A*). Interestingly, these stimuli often produced hyperpolarization and a suppression of spiking in the contralateral DN, suggesting that ipsilateral excitation is accompanied by contralateral inhibition. We also confirmed there was no neural or behavioral response when the *LexA* transgene was omitted (*Figure 5—figure supplement 1*), as expected.

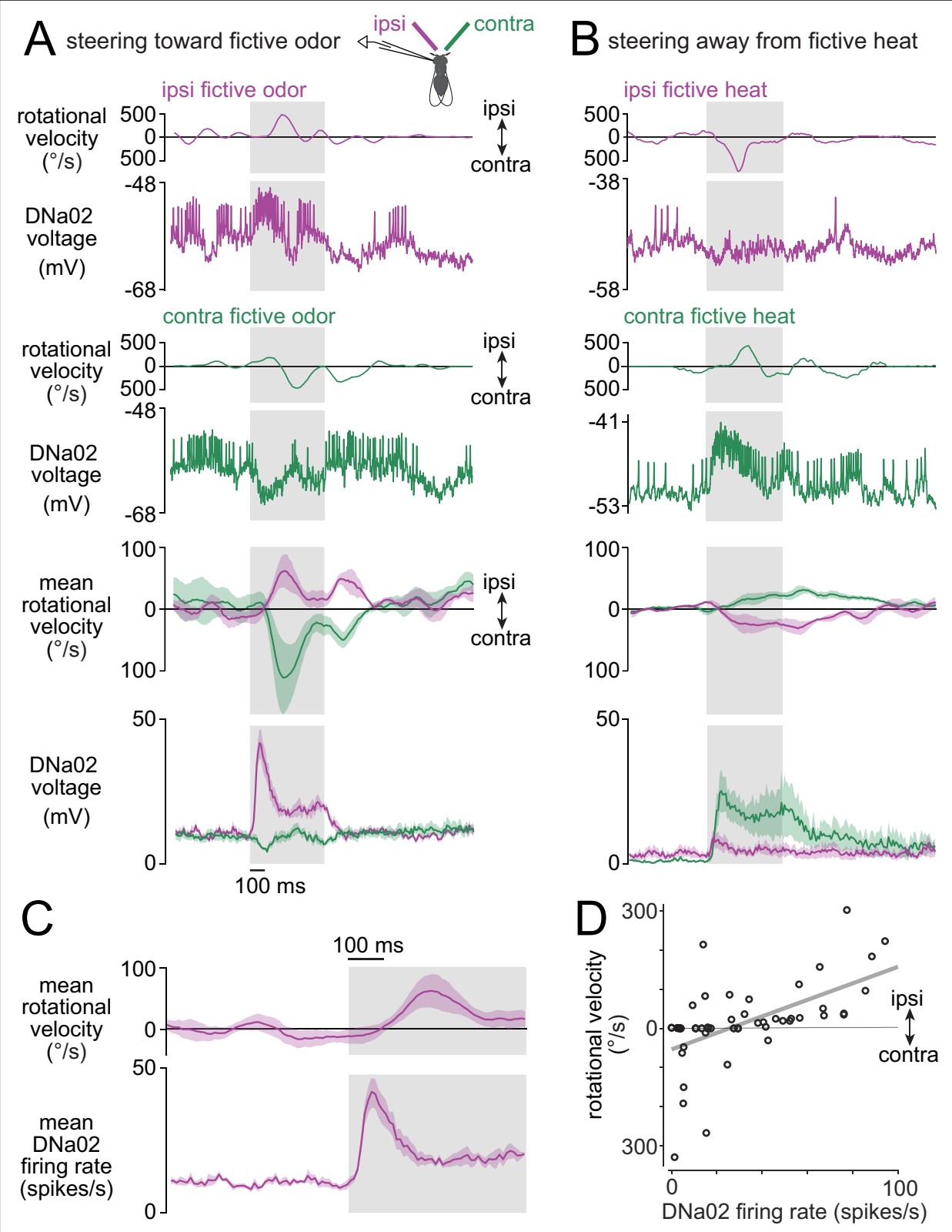

**Figure 5.** DNa02 participates in stimulus-directed steering. (**A**) CsChrimson was expressed in most olfactory receptor neurons. As a fly walked on a spherical treadmill, a fiber-optic filament illuminated either the right or left antennae alternately. Ipsi- and contralateral stimuli are defined relative to the recorded neuron. Top: two example trials, one ipsi and one contra, showing the fly's rotational velocity and DNa02 activity on each trial. Bottom: mean ± SEM across flies, n=4 flies. Gray shading shows the 500 ms period of fictive odor. (**B**) Same as (**A**) but for fictive heat. CsChrimson was expressed in heat-

*Figure 5 continued on next page*

*Figure 5 continued*

activated neurons of the antenna. Fictive heat drives behavioral turning away from the stimulus, rather than toward it. (**C**) Mean data on an expanded timescale to show that DNa02 firing rate increases precede turning toward ipsilateral fictive odor (mean ± SEM across flies, n=4 flies). (**D**) Trial-to-trial variability in an example experiment. Each datapoint is a trial where fictive odor was presented on the ipsilateral side; gray line is linear fit ($R^2$=0.33, p=$10^{-5}$, two-tailed t-test). In other experiments, $R^2$ ranged from 0.16 to 0.40, with p always <0.005.

The online version of this article includes the following figure supplement(s) for figure 5:

**Figure supplement 1.** Genetic controls for fictive sensory stimuli and behavioral responses in intact flies.

**Figure supplement 2.** Lateralized fictive odor stimuli produce asymmetric responses in DNa01.

To generate an aversive stimulus, we expressed CsChrimson in the antennal thermoreceptor neurons that are excited by heat, under the control *Gr28b.d-LexA* (*Frank et al., 2015*). As before, we stimulated the right or left antenna independently. These fictive warm stimuli drove flies to steer away from the stimulated antenna (*Figure 5B*). These stimuli again produced responses in DNa02, but now with higher firing rates on the side contralateral to the stimulus (*Figure 5B*). Thus, DNa02 activity was higher on the side ipsilateral to the attractive stimulus, but contralateral to the aversive stimulus. In other words, DNa02 encoded the laterality of the stimulus-evoked action, not the laterality of the stimulus itself.

On average, DNa02 was activated about 150ms before the stimulus-evoked turn (*Figure 5C*). Moreover, DNa02 could predict much of the trial-to-trial variability in the magnitude of stimulus-evoked turns (*Figure 5D*). Together, these findings suggest that DNa02 participates in reorienting the body in response to a lateralized stimulus. We obtained similar results for DNa01 neurons (*Figure 5—figure supplement 2*), supporting the idea that DNa01 and DNa02 often function together.

Although DNa02 hyperpolarized whenever the fly stopped walking (*Figure 6A and B*), it continued responding to lateralized odors even when the fly was stopped. Indeed, during immobility, the stimulus-evoked change in DNa02 firing rate was similar to what we observed during walking (*Figure 6C and D*). This observation suggests that DNa02 contains information about latent steering drives, even in a quiescent behavioral state where these drives produce no measurable motor consequences. We cannot exclude the possibility that DNa02 is driving postural changes when the fly is stopped, and these postural changes are so small that we cannot detect them. In this case, however, there would still be an interesting mismatch between the stimulus-evoked change in DNa02 firing rate (which is large) and the stimulus-evoked postural response (which would be very small).

## Converging brain pathways onto DNa02

Our results thus far indicate that DNa02 integrates several different types of steering signals. Specifically, it receives internal goal-directed steering signals from the central complex, as well as stimulus-directed steering signals from the olfactory system and the thermosensory system. It also receives locomotor state signals that make the cell more depolarized when the fly is walking. These results motivated us to examine the full-brain connectome (*Dorkenwald et al., 2023*; *Dorkenwald et al., 2022*; *Lin et al., 2024*; *Zheng et al., 2018*) for clues as to how these signals are integrated.

First, we identified DNa02 in the full brain FAFB dataset (*Dorkenwald et al., 2023*; *Dorkenwald et al., 2022*; *Scheffer et al., 2020*; *Schlegel et al., 2023*; *Zheng et al., 2018*); we then proofread all the presynaptic inputs to DNa02, while also matching each cell type with its equivalent in the hemi-brain connectome dataset (*Scheffer et al., 2020*). We contributed all this information (proofreading and cell type names) to the FlyWire community platform (*Dorkenwald et al., 2022*) as this study progressed. The identification of DNa01 is taken from a comprehensive study of all DNs in the FAFB-FlyWire dataset (*Stürner et al., 2024*).

We found that the inputs to DNa02 are largely non-overlapping with those of DNa01 (*Figure 7A*). This fits with our finding that DNa02 can be recruited before DNa01, with a more transient pattern of activity: the inputs to these cells are evidently somewhat independent. Both DNs receive substantial input from central brain intrinsic neurons; however, only DNa02 receives direct input from PFL3 cells (*Figure 7B*, *Figure 7—figure supplement 1*). PFL3 cells compare head direction signals with internal goal signals, and they produce directional steering commands if there is a mismatch between these angular values (*Dan et al., 2024*; *Goulard et al., 2021*; *Hulse et al., 2021*; *Matheson et al., 2022*; *Mussells Pires et al., 2024*; *Rayshubskiy et al., 2020*; *Westeinde et al., 2024*). PFL3 neurons are,

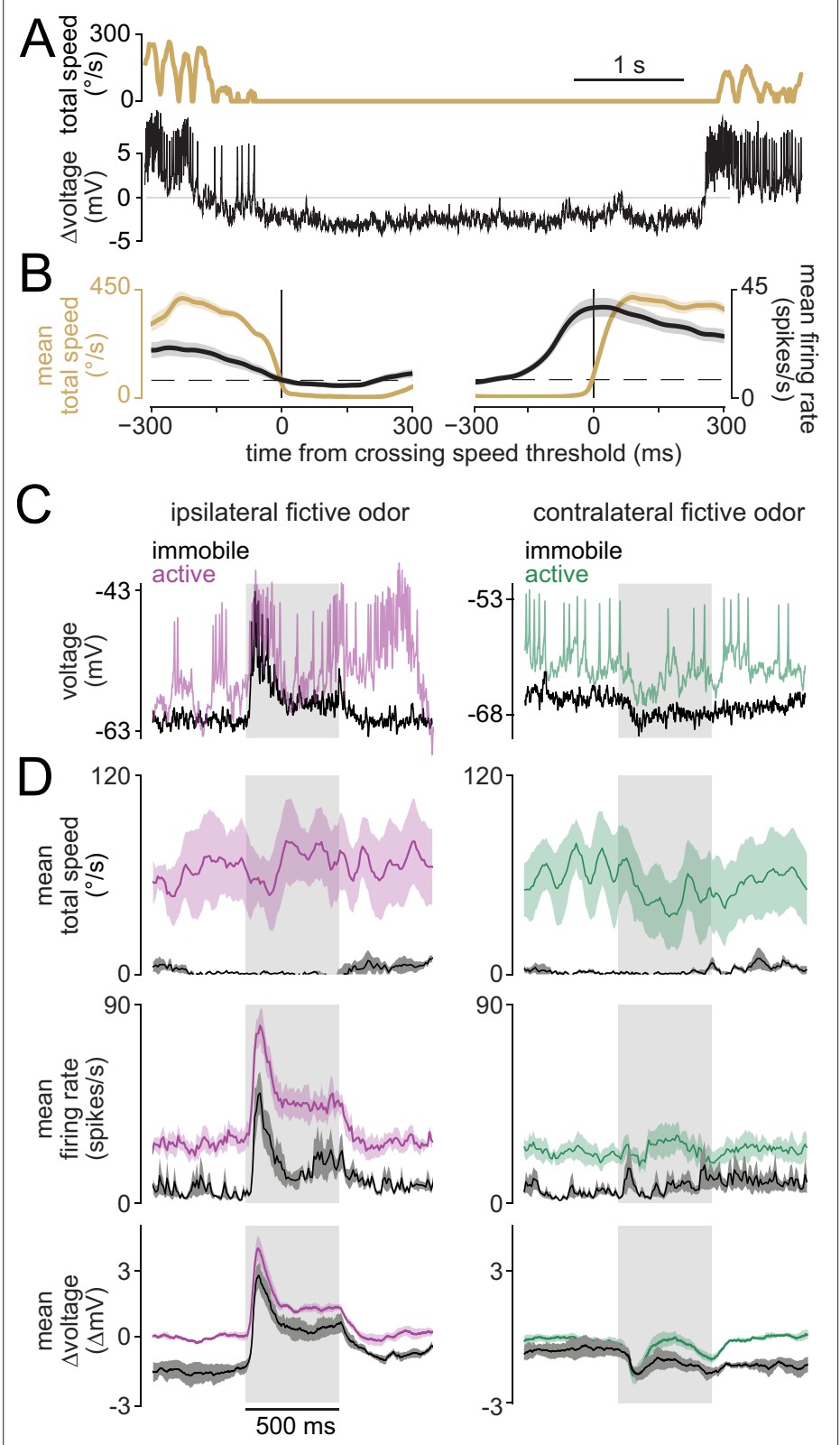

**Figure 6.** Latent steering drives during immobility. (**A**) Example showing how DNa02 hyperpolarizes during immobility. Total speed is defined as rotational speed + sideways speed + forward speed. (**B**) Mean total speed and DNa02 firing rate during (left) the transition to immobility and (right) the transition to activity (± s.e.m. across flies, n=7 flies). Transitions were detected by setting a threshold for total speed (dashed line). The firing

*Figure 6 continued on next page*

*Figure 6 continued*

rate increase precedes the onset of movement by ~250 ms. (**C**) Examples of DNa02 activity during fictive odor presentation when the fly was active versus immobile. Note that, when the fly is immobile, ipsilateral odor still evokes depolarization and spiking. Note also that contralateral odor can still evoke hyperpolarization during immobility. (**D**) Summary data (mean ± s.e.m. across flies, n=4 flies) for active versus immobile trials. Odor produces a similar change in neural activity for trials where the fly is behaviorally active versus immobile. However, the entire dynamic range of neural activity is shifted downward during immobility, so that the peak firing rate (and peak depolarization) is reduced.

therefore, the likely origin of the turning signals we see in DNa02 when we chemogenetically inject an error signal into the head direction system (*Figure 4*).

Notably, however, we found that PFL3 neurons also send many strong indirect projections to DNa02. Specifically, PFL3 neurons project to two cells (DNa03 and LAL010) that make strong excitatory

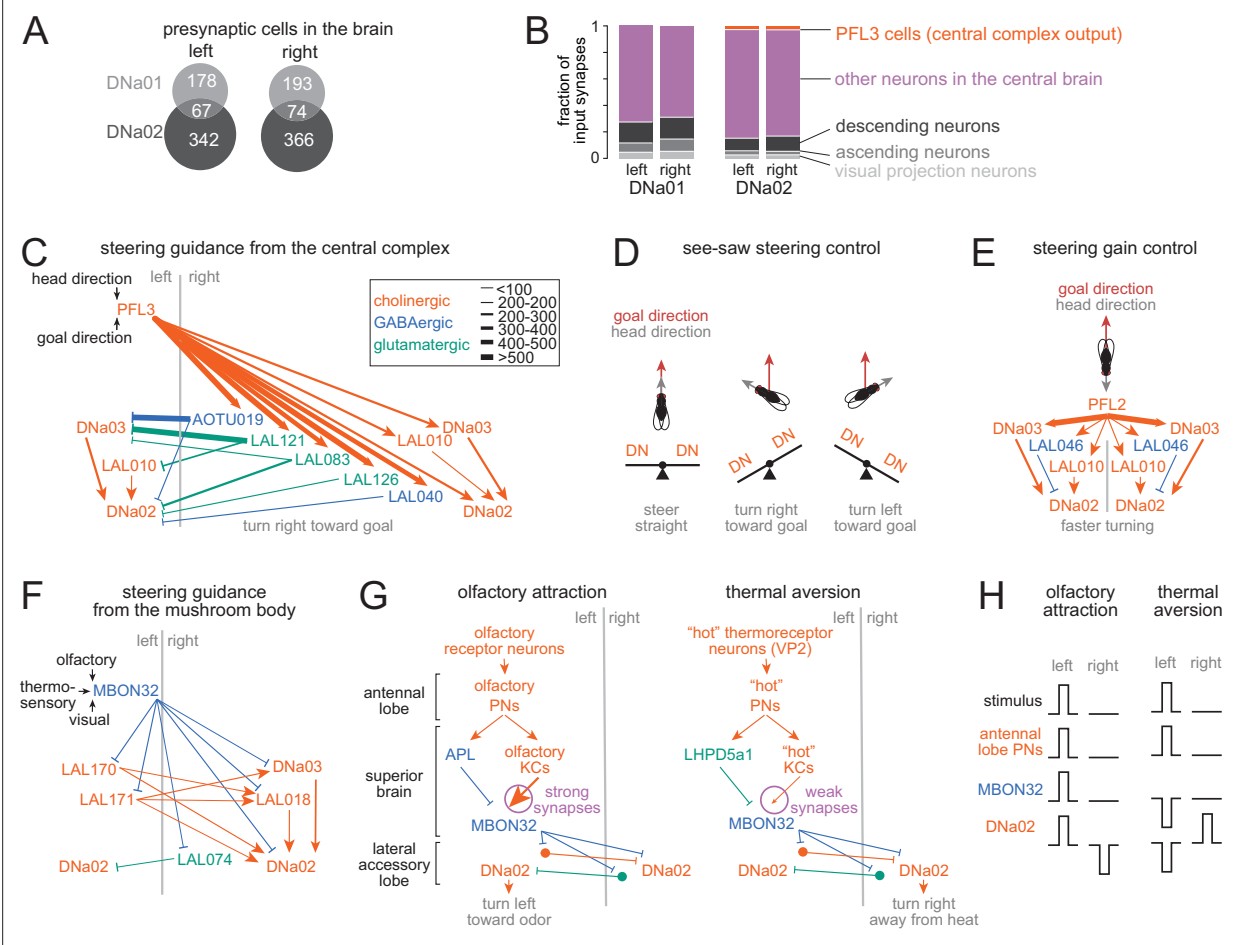

**Figure 7.** Converging brain pathways onto DNa02. (**A**) Number of cells in the brain connectome presynaptic to each descending neuron (DN) (*Dorkenwald et al., 2023*; *Dorkenwald et al., 2022*; *Lin et al., 2024*; *Zheng et al., 2018*). (**B**) Input synapses in the brain, categorized by presynaptic cell type. (**C**) Major pathways from PFL3 cells to DNa02. Glutamatergic connections are likely inhibitory (*Liu and Wilson, 2013*). Line widths denote the number of synapses connecting each cell type (see key). For example, as there are 12 PFL3 cells per hemisphere, their output weights are summed to determine the width of each PFL3 output arrow (here and in E-F). (**D**) Schematic: how PFL3 cells might perform see-saw steering control. When the fly deviates to the left of its goal direction, PFL3 cells are positioned to excite DNa02 on the right, while also inhibiting DNa02 on the left. When the fly deviates to the right of its goal direction, this should occur in reverse. (**E**) Major pathways from PFL2 cells onto DNa02. (**F**) Major pathways from MBON32 cells onto DNa02. (**G–H**) Schematic: how MBON32 might drive steering toward an attractive stimulus (like an attractive odor), as well as steering away from an aversive stimulus (like aversive heat).

The online version of this article includes the following figure supplement(s) for figure 7:

**Figure supplement 1.** Comparing pathways to DNa02 versus DNa01.

ipsilateral connections onto DNa02 (*Figure 7C*). These indirect connections should amplify the influence of central complex steering signals. Moreover, PFL3 neurons also project to several cell types that are positioned to inhibit the contralateral copies of DNa02, DNa03, and LAL010 (*Figure 7C*). Contralateral suppression should increase the right-left difference in DNa02 activity, thereby increasing rotational velocity. In short, central complex projections are wired to have opposite effects on the two copies of DNa02. This architecture suggests a 'see-saw' model of steering, where an increase in excitatory synaptic input in one DN (and/or a disinhibition of that DN) is often accompanied by an increase in inhibitory synaptic input to the contralateral DN (*Figure 7D*). This see-saw model is consistent with our observation that lateralized stimuli can excite one DN while inhibiting the contralateral DN.

That said, some indirect connections onto DNa02 are actually bilaterally symmetric. For example, DNa03 and LAL010 both receive strong bilaterally symmetric projections from another excitatory central complex output cell type, PFL2 (*Figure 7E*). PFL2 neurons are recruited when the fly is oriented away from its internal goal, and direct activation of PFL2 neurons increases the overall gain of steering commands, without specifying the direction of turning (*Westeinde et al., 2024*).

Outside the central complex, there are many pathways for stimulus-directed steering. For example, there are multiple pathways connecting olfactory and thermosensory peripheral cells to DNa02. However, the shortest pathways involve the mushroom body output neuron MBON32 (*Li et al., 2020*). We found that MBON32 projections onto DNa02 are also organized to produce see-saw steering: excitation onto one DN is generally accompanied by inhibition onto the other DN (*Figure 7F*). Interestingly, MBON32 projections could drive turning toward an appetitive stimulus, as well as turning away from an aversive stimulus (*Figure 7G and H*). This idea derives from the fact that there are generally two parallel pathways connecting any set of antennal lobe projection neurons to MBON32: a plastic excitatory pathway, and a fixed inhibitory pathway. In principle, depression of the excitatory pathway (via Kenyon Cells, or KCs) could, therefore, unmask inhibition. Depression of KC→MBON32 synapses should be driven by aversive stimuli, via the dopamine neuron PPL103 (*Cohn et al., 2015*; *Hige et al., 2015*; *Jacob and Waddell, 2020*; *Li et al., 2020*; *Perisse et al., 2016*). Therefore, a stimulus with appetitive associations, such as an appetitive odor, should produce excitation of MBON32, because the active KC→MBON synapses should be strong (not depressed); this would then drive ipsiversive turning. Conversely, a stimulus with aversive associations, such as aversive heat, should produce inhibition of MBON32, because the active KC→MBON synapses should be weak (depressed), which would drive contraversive turning.

Finally, connectome data shows that DNa02 receives abundant visual input. Some of this input comes from a large set of visual projection neurons, which project directly from the optic lobe to DNa02 (*Figure 7A*) as well as DNa03 (*Li et al., 2020*). Another major source of input comes from projection neurons in the anterior optic tubercle, downstream from LC10 cells. LC10 cells are small object motion detectors that mediate steering toward visual objects (*Hindmarsh Sten et al., 2021*; *Ribeiro et al., 2018*; *Schretter et al., 2025*). Some visual signals arrive via MBON32, which receives visual input in addition to olfactory and thermo/hygrosensory input (*Figure 7F*).

In summary, DNa02 is a site of integration for many steering control pathways. These pathways tend to be arranged for see-saw steering control: when one copy of DNa02 is excited, the other is inhibited. Many of these pathways involve parallel excitation and inhibition, which could provide a mechanism to specify steering direction based on excitatory/inhibitory (E/I) balance. When one pathway is plastic, the change in E/I balance could provide a way to convert memory recall into a bidirectional behavioral readout.

## Contributions of single descending neuron types to steering behavior

Finally, we measured the causal influence of DNa02 on steering behavior. To do this, we used hs-FLP to stochastically express the channelrhodopsin variant ReaChR (*Inagaki et al., 2014*) in DNa02 neurons. This produced either unilateral expression or bilateral expression (*Figure 8A*). As each fly walked on a spherical treadmill, we illuminated its head from above to activate the ReaChR +neuron(s). This illumination was symmetric on the two sides of the head.

In the flies with unilateral expression, we found that light evoked a small average steering bias in the ipsilateral direction – i.e., the direction of the ReaChR +neuron (*Figure 8B*). To determine if this steering bias was significant, we used control sibling flies with bilateral ReaChR expression in DNa02. Specifically, each fly with bilateral expression was randomly assigned a label ('right expression' or 'left

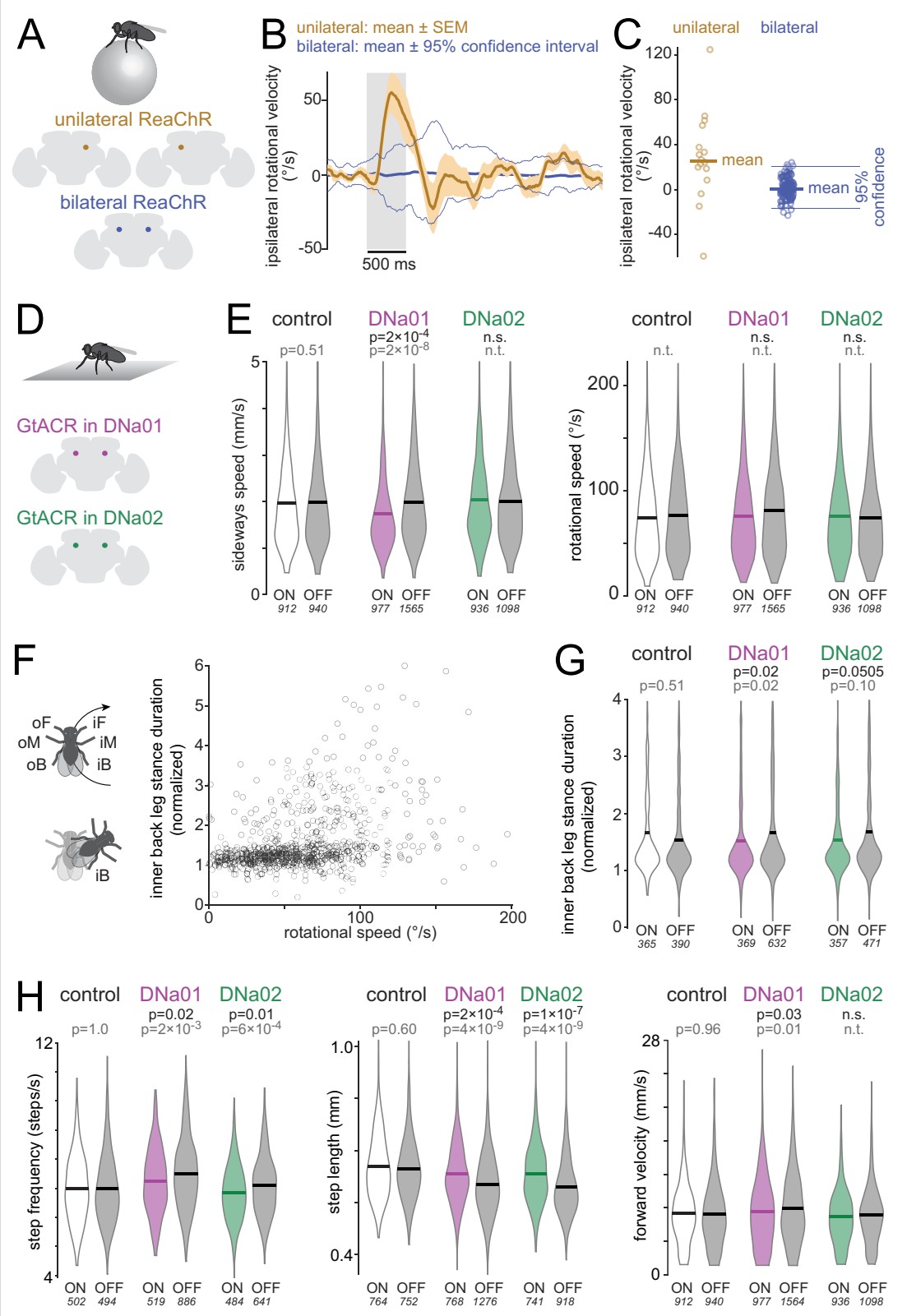

**Figure 8.** Behavioral results of directly activating and silencing descending neurons (DNs). (**A**) As flies walked on a spherical treadmill, they were illuminated repeatedly for 500 ms epochs. ReaChR was expressed uni- or bilaterally in DNa02. (**B**) Ipsilateral rotational velocity, mean of 16/17 flies (uni/bilateral ReaChR). The mean of the unilateral data (thick gold line) lies outside the 95% confidence interval (thin blue lines) of the distribution of outcomes we obtain when we randomly assign bilateral (control) flies to the 'right' or 'left' expression group. (**C**) Ipsilateral rotational velocity. Each

*Figure 8 continued on next page*

*Figure 8 continued*

data point is one fly (or simulated fly), averaged across the illumination epoch. (**D**) GtACR1 was expressed bilaterally in DNa01 or DNa02. Flies were illuminated repeatedly for 2 min epochs as they walked in an arena. (**E**) Sideways and rotational speed distributions for light on/off in each genotype. The p-values in black denote ANOVA genotype ×light interactions after Bonferroni-Holm correction; the p-values in gray denote post hoc Tukey tests comparing light on/off within a genotype; n.s.=not significant, n.t.=not tested (because genotype ×light interaction was not significant). The number of points (500 ms time windows) in each distribution is shown in italics; see Methods for fly numbers. (**F**) Stance duration of inner back (iB) leg, normalized to other legs (oF, oM, oB, iF) was measured in 500 ms time windows (n=755 windows) and plotted versus rotational speed. For some windows where rotational speed is high, iB stance duration is prolonged, i.e., the fly pivots on its iB leg. (**G**) Distribution of normalized iB stance durations, for 500 ms time windows where rotational speed exceeded a threshold of 20 °/s for ≥100 ms. (**H**) Step frequency, step length, and forward velocity distributions.

The online version of this article includes the following figure supplement(s) for figure 8:

**Figure supplement 1.** Leg movements associated with body rotation.

**Figure supplement 2.** Expression of GtACR1::eYFP under the control of descending neuron (DN) split-Gal4 lines.

expression'). Using these flies, we then computed the average rotational velocity in the 'ipsilateral' direction, just as we did for the flies with true unilateral expression. This simulation was repeated independently for 1000 bootstrap replicates (resampling with replacement) to obtain the 95% confidence interval of the mean, under the null hypothesis that there is only a random relationship between DNa02 expression and steering behavior. We found that the mean of the data (with unilateral expression) was outside the 95% confidence interval of the simulation outcomes (*Figure 8B and C*). Thus, in flies with unilateral expression, the turning bias was significantly larger than we would expect by chance.

In these experiments, symmetric illumination from above also caused changes in forward velocity. These effects were likely due to visual startle, because they were indistinguishable in flies with unilateral expression, bilateral expression, and no expression (data not shown). Due to the confounding effect of visual startle, we could not use this dataset to investigate how optogenetic activation of DNa02 affects the fly's forward velocity.

Next, to investigate the effect of bilateral DN inhibition, we expressed an inhibitory opsin (GtACR1) in either DNa01 neurons or DNa02 neurons (*Figure 8D*). As a genetic control, we also used an 'empty' split-Gal4 line (*Hampel et al., 2015*). As each fly walked in a small arena, we switched an overhead light on and off periodically (2 min on, 2 min off). We analyzed behavior as a function of genotype (Gal4/control) and light (on/off) using a set of two-factor ANOVAs, corrected for multiple comparisons. A significant (genotype ×light) interaction would be evidence for a behavioral effect of DN inhibition.

First, we analyzed the body movements that are correlated with DNa01 and DNa02 activity – namely, rotational and sideways movements (*Figure 8E*). When DNa01 neurons were inhibited, sideways speed was significantly reduced. Surprisingly, however, rotational speed was unchanged when either DNa01 or DNa02 neurons were inhibited. Recall that DNa01 and DNa02 predict rotational movements about as well as they predict sideways movements (*Figure 1F*). Moreover, unilateral activation of DNa02 influences ipsilateral rotational movements (*Figure 8B and C*). Why, then, is the body's rotational movement unaltered when these DNs are bilaterally inhibited? In freely walking flies, body rotation can be achieved via many different leg maneuvers (*DeAngelis et al., 2019*; *Katsov et al., 2017*; *Yang et al., 2024*). For example, in some body-rotation events, the stance duration of the inner back leg is prolonged relative to the other legs – in essence, the fly pivots on its inner back leg – whereas this maneuver is absent in other body-rotation events (*Figure 8F*). This raised the possibility that DN hyperpolarization interferes with a specific leg maneuver, rather than body rotation per se.

To investigate this idea, we identified five leg maneuvers that often accompany body rotations in freely walking control flies (*Figure 8—figure supplement 1*). We then examined whether each leg maneuver was altered by DN inhibition. We found that inhibiting DNa01 produced a significant defect in stance prolongation in the inner back leg – i.e., it reduced the fly's tendency to pivot on the inner back leg during a body rotation (*Figure 8G*). We observed the same trend when we inhibited DNa02, although here the trend fell short of statistical significance. These results suggest that inhibiting DNa01 (and possibly DNa02) causes a defect in inner-back-leg-stance-prolongation, but this defect is evidently compensated via other mechanisms to achieve a normal rotational movement of the body in freely moving flies. We also noticed that inhibiting either DNa01 or DNa02 altered the fly's normal leg movement rhythm, regardless of whether the fly was walking straight or turning. Specifically, step

length increased, while step frequency decreased (*Figure 8H*). Inhibiting DNa01 also decreased the fly's forward velocity (*Figure 8H*).

All these effects are weak, and so they should be interpreted with caution. Also, both DN split-Gal4 lines drive expression in a few off-target cell types, which is another reason for caution (*Figure 8— figure supplement 2*). That said, these results suggest that both DNs can lengthen the stance phase of the ipsilateral back leg, which would promote ipsiversive turning. These results are also compatible with a scenario where both DNs decrease the step length in the ipsilateral legs, which would also promote ipsiversive turning. Step frequency does not normally change asymmetrically during turning, so the observed decrease in step frequency during optogenetic inhibition may just be a by-product of increasing step length when these DNs are inhibited.

## Discussion
### Ensemble codes for steering

In this study, we showed that DNa01 and DNa02 are both recruited during turning bouts in walking flies. However, connectome data shows these two DNs receive largely non-overlapping inputs in the brain, and they have largely non-overlapping outputs in the ventral nerve cord, suggesting that they represent parallel and largely independent pathways for steering control. Importantly, our electrophysiological results in this study support this hypothesis: these two DN types have distinct functional profiles: DNa02 predicts large, rapid steering events, while DNa01 predicts smaller and slower steering events. Moreover, DNa02 predicts more transient changes in steering, as compared to DNa01. At the onset of a steering bout, DNa02 is recruited earlier, and its firing rate fluctuations are more transient than those of DNa01. All these results argue that these two DNs play specialized roles.

Our study does not fully answer the question of how these DNs affect leg kinematics, because we were not able to simultaneously measure DN activity and leg movement. However, our optogenetic experiments suggest that both DNs can lengthen the stance phase of the ipsilateral back leg (*Figure 8G*), and/or decrease the step length in the ipsilateral legs (*Figure 8H*), either of which would promote ipsiversive turning. If these DNs have similar qualitative effects on leg kinematics, then why does DNa02 precede larger and more rapid steering events? This may be due to the fact that DNa02 receives stronger and more direct input from some steering circuits in the brain (*Figure 7—figure supplement 1*). It may also relate to the fact that DNa02 has more direct connections onto motor neurons (*Figure 1B*).

These subsequent studies also confirmed some of our key results regarding DNa02 and DNa01. Specifically, *Yang et al., 2023* and *Feng et al., 2024* tested the effect of unilateral excitation of DNa02 (using a different genetic strategy), and found that this produced ipsilateral body rotation, consistent with our results. *Feng et al., 2024* also confirmed that bilateral ablation of DNa02 significantly reduced the velocity of corrective turning maneuvers induced by errors in the head direction system, and this supports our conclusion that DNa02 is recruited by the central complex for control of head direction. Meanwhile, *Braun et al., 2024* reported that bilateral optogenetic activation of DNa02 or DNa01 can drive turning in either direction.

Interestingly, *Yang et al., 2023* found that perturbing DNa02 mainly affected ipsilateral leg movements, whereas (during voluntary turns) DNa02 activity was actually correlated with both ipsi- and contra legs, with opposite changes in the step cycle on the two sides of the body (*Yang et al., 2023*). Our brain connectome analyses in this study actually explain this effect. Specifically, we show that the brain inputs onto DNa02 are arranged so as to produce concurrent excitation of one copy and inhibition of the other copy, a see-saw arrangement that should generate opposite leg movement changes on the two sides of the body.

*Yang et al., 2023* found that unilateral inhibition of DNa02 had no measurable effect on body rotation, but it significantly increased ipsilateral step length. This is reminiscent of our finding that bilateral DNa02 inhibition decreases step length bilaterally (as does bilateral inhibition of DNa01). *Yang et al., 2023* this may be due to the fact that the latter study focused on flies walking a spherical treadmill, whereas our perturbation experiments focused on more naturalistic leg movements in freely walking flies.

Could these 'steering DNs' also play a role in the control of the fly's forward motion during straight-line walking? A previous study reported that optogenetic activation of either DNa01 or DNa02

increases global locomotor activity (*Cande et al., 2018*). In this study, we found that bilaterally opto-genetically inhibiting DNa01 decreased the fly's forward velocity. However, our electrophysiological results show that DNa01 and DNa02 are only weakly correlated with forward velocity during normal locomotion. In short, it seems that DNa01 and DNa02 are normally recruited by the brain for steering control, but bilateral perturbations can nonetheless produce forward velocity changes. It is perhaps not surprising that bilaterally perturbing these DNs can change forward velocity, as these DNs can only drive steering via laterally asymmetric changes in the step cycle (*Cheong et al., 2023*; *DeAngelis et al., 2019*; *Feng et al., 2024*; *Strauss and Heisenberg, 1990*; *Yang et al., 2023*), and any imposed bilateral change in the step cycle will almost inevitably affect the fly's forward velocity. That said, it remains possible that DNa01 and DNa02 contribute to modulations of forward velocity during straight-line locomotion under some specific conditions.

Finally, there is emerging evidence that many DNs actually influence steering during walking. We found that optogenetically inhibiting DNa01 produced only small defects in steering, and inhibiting DNa02 did not produce statistically significant effects on steering; these results make sense if DNa02 is just one of many steering DNs. Indeed, while this study was in revision, several subsequent studies identified additional steering DNs, including DNg13, DNb05, and DNb06 (*Yang et al., 2023*), DNa11, DNae003, DNa03, and DNae014 (*Feng et al., 2024*), and DNb02 (*Braun et al., 2024*). Several subsequent studies have also provided evidence that some DNs involved in steering control are organized hierarchically, with specific 'broadcaster' DNs subsequently recruiting other DNs with more specialized functions (*Braun et al., 2024*; *Sapkal et al., 2024*). For example, DNa02 is downstream from DNa03, a broadcaster DN that seems to promote decreases in forward speed as well as ipsiversive steering (*Feng et al., 2024*; *Westeinde et al., 2024*); DNa02 is also downstream from DNp09, a broadcaster DN that can promote increases in forward speed as well as ipsiversive steering (*Bidaye et al., 2020*; *Braun et al., 2024*; *Sapkal et al., 2024*).

## Behavioral state and latent motor commands

The DNs that target the wings and the legs are largely distinct populations (*Namiki et al., 2018*), and the DNs currently implicated in steering during flight are distinct from those we have identified here (*Ros et al., 2024*; *Schnell et al., 2017*). However, DNa02 does have indirect connectivity onto wing motor neurons, suggesting it contributes to steering during flight (*Cheong et al., 2023*). Moreover, a study subsequent to this work reported that DNa03 activation can promote turning in flight, as well as turning during walking (*Feng et al., 2024*). Interestingly, the GABAergic cell VES041 can suppress steering during flight (*Ros et al., 2024*), and we predict that it has a similar role during walking, based on its connectivity upstream from DNa02. It will be interesting to learn how much upstream brain circuitry is shared for flight control and walking control, and what happens to each of these systems during the behavioral state when they are not currently in use.

In this regard, it is notable that we observed clear responses to lateralized sensory stimuli in steering DNs even when the fly was not walking. Similarly, a recent study found that *Drosophila* DNs that evoke landing during flight are still responsive to visual stimuli when the fly is not flying (*Ache et al., 2019*). We might think of these residual DN responses as a latent bias toward action, or a preparation for an action that never ultimately occurs (*Kien, 1990*). In this regard, it is relevant that mammalian cortico-spinal neurons can be active in the absence of movement, either in the period preceding a movement (*Tanji and Evarts, 1976*) or when a movement is being simply observed (*Kraskov et al., 2014*). These latent corticospinal signals may represent movement preparation, which may additionally involve neurons which are not actually active during movement (*Churchland et al., 2010*), and they may be responsible for driving latent activity in spinal neurons in the absence of movement (*Prut and Fetz, 1999*). The discovery of latent motor activity in the *Drosophila* brain provides a potential starting point for future mechanistic investigations of brain dynamics during motor preparation.

Behavioral state transitions may be organized, at least in part, via the network of cells presynaptic to DNs ('pre-DNs') in the lateral accessory lobe (*Namiki and Kanzaki, 2016*). Our analyses highlight the anatomical features of several pre-DNs, most notably DNa03 and LAL010, which both receive bilateral input from cell types known to exert high-level premotor control, namely PFL2 (*Westeinde et al., 2024*) and VES041 (*Ros et al., 2024*). Another pre-DN with similar connectivity (LAL013) has recently been shown to play an important role in modulating steering (*Feng et al., 2024*).

## From flies to vertebrates

Our work shows that steering is driven by combinations of DNs with distinct functional roles, recruited by distinct parallel pathways in the brain, and targeting largely non-overlapping sets of cells in the ventral cord. These results are relevant to vertebrates, because any walking or running organism faces the same basic physical constraints (*Dickinson et al., 2000*), and so it is likely that evolution has found a common set of solutions to these basic problems. In vertebrates, recent work has begun to identify the DNs that influence steering (*Cregg et al., 2020*; *Usseglio et al., 2020*). Based on our results, we would predict that vertebrate steering DNs should also be diverse and specialized.

In all species, orienting behaviors are fundamental and ubiquitous indicators of an organism's perceptions and intentions. As such, they provide a useful window into perception and cognition. Once we fully understand the specializations of the DNs that control steering and orienting, we should be able to reframe the problem of controlling an organism's orientation as a more concrete problem of generating a specific pattern of DN dynamics. This motor-centric perspective then allows us to understand sensory and cognitive computations as solutions to specific problems of dynamical system control.*Figure 3*.

## Methods
### Fly husbandry and genotypes

We used female flies 1–2 d post-eclosion, except where specified below. We used virgins to minimize egg-laying, which tended to stop the movement of the spherical treadmill. Flies were housed in a 25° incubator at 50–70% relative humidity. Unless otherwise noted, flies were cultured on molasses food (B7 recipe, Archon Scientific) and kept on a 12/12 light/dark cycle. Flies used in optogenetic experiments were cultured on molasses food and then transferred 0–1 d after eclosion onto a vial of standard cornmeal agar fly food supplemented with rehydrated potato flakes (Formula 4–24, Carolina Biological Supply) and 100 µL of all-*trans*-retinal (35 mM in ethanol; R2500, Sigma-Aldrich). These vials were covered with aluminum foil to prevent photoconversion of all-*trans*-retinal by ambient light.

The split-Gal4 lines (*Pfeiffer et al., 2010*) targeting DNs (*Namiki et al., 2018*) were *SS00730* for DNa02 (*R75C10-p65ADZp* in *attP40*; *R87D07-ZpGdbd* in *attP2*, RRID:BDSC_88574) and *SS00731* for DNa01 (*R22C05-p65ADZp* in *attP40*; *R56G08-ZpGdbd* in *attP2*, RRID:BDSC_75862); these were obtained from Gwyneth Card (Janelia Research Campus). Each of these split-Gal4 lines typically drives expression in one DN per brain hemisphere, although *SS00731* occasionally drives expression in two DNs per hemisphere. Images of these split-Gal4 lines are available from the FlyLight split-Gal4 image database (https://splitgal4.janelia.org/cgi-bin/splitgal4.cgi). Both lines are relatively specific for the DNs in question, with limited and reproducible patterns of off-target expression (*Figure 8—figure supplement 2*).

*R75C10-Gal4(attP2)*, RRID:BDSC_48322 (*Jenett et al., 2012*; *Pfeiffer et al., 2008*) and *R60D05-LexA(attP40)*, RRID:BDSC_52867 (*Pfeiffer et al., 2010*) were obtained from the Bloomington *Drosophila* Stock Center (BDSC). Images of these lines are available from the FlyLight 'Generation 1' image database (https://flweb.janelia.org/). *VT032906-Gal4(attP2)* (*Tirian and Dickson, 2017*) was obtained from the Vienna *Drosophila* Resource Center. An image of this line is available from the Virtual Fly Brain database (https://v2.virtualflybrain.org/).

*Orco-LexA* (on chromosome 3) (*Lai and Lee, 2006*) was obtained from Tzumin Lee (Janelia Research Campus). *Gr28b.d-LexA* (on chromosome 2) (*Frank et al., 2015*) was obtained from Marco Gallio (Northwestern University). *hsFLP.122* ('Bloomington Drosophila Stock Center. Notes on X-linked insertions of heat shock-FLP constructs,' n.d.; *Chou and Perrimon, 1996*; *Golic and Lindquist, 1989*) was obtained from Thomas Clandinin (Stanford University). *UAS-GtACR1::eYFP(VK00005)* (*Mauss et al., 2017*) was obtained from Michael Crickmore (Boston Children's Hospital). *pJFRC81-td3-Halo7::CAAX(attp18)* was obtained from Greg Jefferis via Luke Lavis; this specific reagent has not been published previously, although general methods of its construction have been published (*Sutcliffe et al., 2017*). *UAS-P2X2* (chromosome 3) (*Lima and Miesenböck, 2005*) was obtained from Gero Miesenböck (Oxford University).

Finally, the following stocks were obtained from the BDSC: *10XUAS-IVS-mCD8::GFP(su(Hw)attP8)*, RRID:BDSC_32189 (*Pfeiffer et al., 2010*), *10XUAS-IVS-mCD8::GFP(attP40)*, RRID:BDSC_32186 (*Pfeiffer et al., 2010*), *UAS(FRT.mCherry)ReaChR::citrine(VK00005)*, RRID:BDSC_53740 (*Inagaki*

*et al., 2014*), *p65.AD.Uw(attP40); GAL4.DBD.Uw(attP2)*, RRID:BDSC_79603 (*Hampel et al., 2015*), *13XLexAop2-IVS-GCaMP6f-p10(su(Hw)attP5)*, RRID:BDSC_44277 (*Chen et al., 2013*), *13XLexAop2-IVS-CsChrimson::mVenus(attP40)*, RRID:BDSC_55138 (*Klapoetke et al., 2014*), *13XLexAop2-IVS-CsChrimson::mVenus(attP18)*, RRID:BDSC_55137 (*Klapoetke et al., 2014*) *13XLexAop2-IVS-CsChrimson::mVenus(attP2)*, RRID:BDSC_55139 (*Klapoetke et al., 2014*).

Genotypes of fly stocks used in each figure are as follows:

*DNa02 single/dual recording (Figures 1 and 3; Figure 1—figure supplement 2, Figure 3—figure supplement 1)*
y$^1$, w, 10XUAS-IVS-mCD8::GFP(su(Hw)attP8) / w; R75C10-p65ADZp (attP40) / +; R87D07-ZpGdbd (attP2) / + and
w$^{1118}$, 13XLexAop2-IVS-CsChrimson::mVenus(attP18) / w; 10XUAS-IVS-mCD8::GFP(attP40) / R75C10-P65ADZP(attP40); Orco-LexA / R87D07-ZpGdbd(attP2)

*DNa01 single/dual recording (Figure 1; Figure 1—figure supplement 2, Figure 3—figure supplement 2)*
y$^1$, w, 10XUAS-IVS-mCD8::GFP(su(Hw)attP8) / w; R22C05-p65ADZp(attP40) / +; R56G08-ZpGdbd(attP2) / +

*DNa01/DNa02 dual recording (Figure 2; Figure 1—figure supplement 1)*
y$^1$, w, 10XUAS-IVS-mCD8::GFP(su(Hw)attP8) / w; R22C05-p65ADZp(attP40) / 10XUAS-IVS-mCD8::GFP(attP40); R56G08-ZpGdbd(attP2) / R75C10-GAL4(attP2)

*Fictive odor with DNa02 recording (Figure 5; Figure 5—figure supplement 1)*
w$^{1118}$, 13XLexAop2-IVS-CsChrimson::mVenus(attP18) / w; 10XUAS-IVS-mCD8::GFP(attP40) / R75C10-P65ADZP(attP40); Orco-LexA / R87D07-ZpGdbd(attP2)

*Fictive heat with DNa02 recording (Figure 5; Figure 5—figure supplement 1)*
y$^1$, w, 10XUAS-IVS-mCD8::GFP(su(Hw)attP8) / w; R75C10-P65ADZP(attP40) / Gr28b.d-LexA; R87D07-ZpGdbd(attP2) / 13XLexAop2-IVS-CsChrimson::mVenus(attP2)

*Fictive odor with DNa01 recording (Figure 5—figure supplement 2)*
y$^1$, w, 10XUAS-IVS-mCD8::GFP(su(Hw)attP8) / w; R22C05-p65ADZp(attP40) / Orco-LexA; R56G08-ZpGdbd(attP2) / 13XLexAop2-IVS-CsChrimson::mVenus(attP2)

*No-LexA control for fictive odor and fictive heat, DNa02 recordings (Figure 5—figure supplement 1)*
y$^1$, w, 10XUAS-IVS-mCD8::GFP(su(Hw)attP8) / w; R75C10-P65ADZP(attP40) / +; R87D07-ZpGdbd(attP2) / 13XLexAop2-IVS-CsChrimson::mVenus(attP2)

*No-LexA control for fictive odor and fictive heat, DNa01 recordings (Figure 5—figure supplement 2)*
y$^1$, w, 10XUAS-IVS-mCD8::GFP(su(Hw)attP8) / w; R22C05-p65ADZp(attP40) / +; R56G08-ZpGdbd(attP2) / 13XLexAop2-IVS-CsChrimson::mVenus(attP2)

*Calcium imaging and DNa02 recording during central complex stimulation (Figure 4)*
w, pJFRC81-td3-Halo7::CAAX(attP18) / w; R75C10-P65ADZP(attP40) / R60D05-LexA(attP40), 13XLexAop2-IVS-GCaMP6f-p10(su(Hw)attP5); R87D07-ZpGdbd(attP2) / UAS-P2X2, VT032906-Gal4(attP2)

*Central complex-evoked turning, genetic control lacking PEN1 Gal4 (Figure 4)*
w, pJFRC81-td3-Halo7::CAAX(attP18) / w; R75C10-P65ADZP(attP40) / R60D05-LexA(attP40), 13XLexAop2-IVS-GCaMP6f-p10(su(Hw)attP5); R87D07-ZpGdbd(attP2) / UAS-P2X2

*Unilateral optogenetic activation of DNa02 (Figure 8)*
y, w, hsFLP.122 / w; R75C10-P65ADZP(attP40) / +; R87D07-ZpGdbd(attP2) / UAS-FRT.mCherry.FRT.ReachR::citrine(VK00005)

*Bilateral optogenetic silencing of DNa01 (Figure 8; Figure 8—figure supplements 1 and 2)*
w; R22C05-p65ADZp(attP40) / +; R56G08-ZpGdbd(attP2) / UAS-GtACR1::eYFP(VK00005)

*Bilateral optogenetic silencing of DNa02 (Figure 8; Figure 8—figure supplements 1 and 2)*
w; R75C10-P65ADZP(attP40) / +; R87D07-ZpGdbd(attP2) / UAS-GtACR1::eYFP(VK00005)

*Genetic control for optogenetic silencing ('empty' split-Gal4) (Figure 8; Figure 8—figure supplement 1)*
w; p65.AD.Uw(attP40) / +; GAL4.DBD.Uw(attP2) / UAS-GtACR1::eYFP(VK00005)

### Sample size

Sample sizes were chosen based on conventions in our field for standard sample sizes. These sample sizes are conventionally determined on the basis of the expected magnitude of animal-to-animal variability, given published results and pilot data.

### Fly mounting and dissection for electrophysiology and calcium imaging

On the day of the experiment, a fly was cold-anesthetized and then inserted into the hole of a machine-milled (Harvard Medical School Research Instrumentation Core) or photoetched (Etchit) platform made of stainless steel shim stock (McMaster-Carr). We secured the fly to the platform using UV-curable glue (AA3972 Loctite) applied on the thorax and eyes.

We then extended the proboscis with forceps and waxed it in place to prevent brain motion (Electra Waxer, Almore); prior to this step, a manual manipulator (MX160R Siskiyou) was used to lower a shield over the legs to protect them from the tip of the waxer. We then covered the brain with extracellular saline composed of: 103 mM NaCl, 3 mM KCl, 5 mM TES, 8 mM trehalose, 10 mM glucose, 26 mM $NaHCO_3$, 1 mM $NaH_2PO_4$, 1.5 mM $CaCl_2$, and 4 mM $MgCl_2$ (osmolarity 270–275 mOsm). The saline was bubbled during the dissection with 95% $O_2$ and 5% $CO_2$ and reached an equilibrium pH of 7.3. The top of the cuticle was removed, followed by air sacs and fat globules around the patching site. Muscle 16 was severed to prevent brain motion. The frontal head air sac, which extends ventrally under the brain, was also removed to improve illumination from below the brain. Finally, we mechanically removed the peri-neural sheath over the patching site. (For behavior-only experiments in *Figure 8*, *Figure 8—figure supplement 1*, all these dissection steps were omitted.)

Finally, we transferred the fly to the spherical treadmill, where we used a motorized three-axis manipulator (MT3-Z8, Thorlabs) to adjust the fly relative to the ball. Two cameras (BFLY-PGE-13E4M, FLIR) were used to visualize the fly during this adjustment.

### Spherical treadmill and behavioral measurements

We used a sensor-based setup for tracking the movements of the spherical treadmill (*Seelig et al., 2010*). Briefly, a hollow HDPE sphere (6.35 mm diameter) was floated on pressurized air, in a plenum mounted on a manipulator to allow alignment of the center of the ball with two motion sensors located at a distance from the ball. The ball was painted with a random dot pattern to make it easier for motion sensors to detect movement. We positioned two cameras 90° apart to track the motion of the ball in three dimensions (rotational, sideways, and forward velocity), and we took the fly's fictive movements in each direction as equal and opposite to the ball's movements. These motion cameras were custom-built by housing the motion sensor array (ADNS-9800, JACK Enterprises) in a custom housing with a lens (M2514-MP2, Computar). Data from the sensor was read out and translated into an analog signal using a digital-to-analog converter (MCP4725, SparkFun) and an Arduino device (Due, Arduino). Analog data were digitized at 10 kHz with a multi-channel NI DAQ board (PCIe-6353, National Instruments), controlled via the Matlab Data Acquisition Toolbox interface (MathWorks). Analog data were then smoothed using a second-order lowpass Butterworth filter and downsampled to 100 Hz.

Motion sensor outputs were calibrated to obtain the conversion from sensor outputs to ball velocities. We performed this calibration by rotating a ball using a planetary gear motor with an encoder (Actobotics, Part 3638298). The motor was rotated at six different speeds in all three axes of motion, and linear regression was used to calculate the mapping between the output of the motion sensors and the rotation of the ball.

We noticed that steering responses to sensory stimuli were larger and more consistent in intact flies, as compared to flies used for electrophysiology experiments (*Figure 5—figure supplement 1*). The reduced performance of flies used for electrophysiology experiments is likely due to local removal of the peri-neural sheath and/or local disruption of the neuropil surrounding the targeted somata. Nonetheless, the qualitative features of behavior are comparable in intact flies (*Figure 5—figure supplement 1*) versus flies used for electrophysiology (*Figure 5*).

### Electrophysiology

For patch clamp experiments, we illuminated the brain with an infrared fiber (SXF30-850, Smart Vision Lights) mounted under the fly tethering platform in a sagittal plane at ~20° from the horizontal, and

we visualized the brain with a camera (BFLY-PGE-13E4M, Point Gray) mounted on a fluorescence microscope (BX51, Olympus).

Before attempting to obtain seals, we first used a large-diameter glass pipette filled with extracellular saline (4–5 MΩ), to clear the area around the soma of interest by applying positive pressure and also gently sucking away any cells lying on top of the soma of interest. Patch pipettes (9–11 MΩ) were pulled the day of the recording and filled with internal solution containing 140 mM KOH, 140 mM aspartic acid, 10 mM HEPES, 1 mM EGTA, 1 mM KCl, 4 mM MgATP, 0.5 mM Na$_3$GTP, and 13 mM biocytin hydrazide (pH adjusted to 7.2±0.1, osmolarity adjusted to 265±3 mOsm). Whole-cell patch-clamp recordings were performed from fluorescently labeled somata in current-clamp mode using an Axopatch 200B amplifier. Data were low-pass filtered at 5 kHz, digitized at 10 kHz by a 16-bit A/D converter (National Instruments, BNC 2090-A), and acquired using the Matlab Data Acquisition Toolbox (MathWorks). Recordings were stable for at least 15 min and up to 2 hr.

We always recorded from the neuron in the left hemisphere, except when we recorded from the same cell type bilaterally. For bilateral recordings, we performed the cleaning and air-sac clearing steps on both sides of the brain. For dual recording experiments of DNa01 and DNa02, we performed recordings on the left side of the brain.

We targeted DNa01 using *SS00731*, and we targeted DNa02 using *SS00730* (*Namiki et al., 2018*); the only exception was that, in dual DNa01 and DNa02 recordings, we targeted DNa01 using the *SS00731*, and we targeted DNa02 using *R75C10-Gal4*. In these flies, we could identify DNa01 and DNa02 based on the depths of their somata (DNa02 is more superficial), the sizes of their somata (DNa02 is larger), and their spike waveforms (DNa02 spikes are larger). Finally, to confirm the discriminability of DNa02 and DNa01 spikes, we quantified the performance of automated spike waveform classification (*Figure 1—figure supplement 1*).

## Calcium imaging and electrophysiology during central complex stimulation

In the experiments for *Figure 4*, the pitch angle of the fly's head was carefully adjusted during the dissection procedure. This allowed patch-pipette access to the anterior side of the brain (where DNa02 somata reside) as well as pressure-pipette access to the protocerebral bridge on the posterior side of the brain (where PEN1 dendrites reside).

To position the pressure-ejection pipette in the protocerebral bridge and to obtain a patch recording from an DNa02 somata, we visualized the brain and the patch pipette using an infrared fiber (SXF30-850, Smart Vision Lights) mounted under the tethering platform in a sagittal plane at ~20° from the horizontal and a near-infrared camera (GS-U3-41C6NIR FLIR) mounted on the widefield viewing port of a 2-photon microscope (Bergamo II, Thorlabs).

To obtain a recording from DNa02, we needed to fluorescently label DNa02 somata. However, we needed to label DNa02 without also labeling PEN1 neurons (which also expressed Gal4), because PEN1 labeling could have contaminated our GCaMP6f signal in the ellipsoid body. We, therefore, used a chemogenetic approach (*Sutcliffe et al., 2017*). Specifically, we drove an expression of HaloTag under *Gal4/UAS* control, and we then applied SiR110-HaloTag dye (*Zheng et al., 2019*) (a gift from Luke Lavis) to the preparation. The dye was prepared as a stock solution (500 µM in DMSO), and then 1 µL of the stock was dissolved in 500 µL of extracellular saline. Most of the saline in the recording chamber was removed and the dye solution was added to the chamber. After a 15 min incubation, the chamber was rinsed and refilled with regular extracellular saline. We used a Texas Red filter cube (49017, Chroma) to visualize labeled DNa02 somata. Because the perineural sheath was removed selectively in the vicinity of DNa02 somata, the bath-applied SiR110-HaloTag dye bound mainly to DNa02 somata, with essentially no binding to the ellipsoid body neuropil.

For GCaMP6f imaging of EPG neuron dendrites in the ellipsoid body, we used a volumetric, galvo-resonant scanning 2-photon microscope (Bergamo II, Thorlabs) equipped with a 20x, 1.0 n.a. objective (XLUMPLFLN20XW, Olympus) and GaAsP detectors (Hamamatsu). We used ScanImage 2018 software (Vidrio Technologies) to control the microscope. Two-photon excitation was provided by a Chameleon Ultra II Ti:Sapphire femtosecond pulsed laser with pre-compensation (Vision-S, Coherent). To image GCaMP6f, we tuned the laser to 940 nm. The objective was translated vertically using a scanner with a 400 µm travel range (P-725KHDS PIFOC, Physik Instrumente). The ellipsoid body was volumetrically imaged with 12 *z*-Planes of 256×64 pixels separated by 4–5 µm. We acquired ~12 vol/s.

For PEN1 stimulation, we followed a published procedure (*Green et al., 2019*). We prepared a solution of ATP (A7699, Sigma, 0.5 mM) and Alexa594 dye (A33082, FisherScientific, 20 µM) in extracellular saline. This solution was used to fill a glass pipette slightly smaller than a patch pipette. We used a red emission filter cube (49017, Chroma) to visually locate the pipette via the microscope eyepieces. The tip of the pipette was placed into the protocerebral bridge, where the dendrites of PEN1 neurons are located, generally close to glomerulus 4 (*Wolff et al., 2015*). To eject ATP and dye, we delivered a 50–100 ms pulse of pressure to the back of the pipette (10–20 p.s.i.) using a pneumatic device gated by a voltage pulse (PV820; World Precision Instruments). We confirmed ejection by observing a bolus of dye appear in the center of the protocerebral bridge neuropil, and the resulting stimulus-locked rotation of the EPG bump in the ellipsoid body.

In a given experiment, we noticed that the bump tended to jump to the same location in every trial. We used that fact to try to obtain large bump jumps. Specifically, we manually timed the ATP injection in each trial to maximize the size of the resulting bump jump. We ejected ATP every 15–25 s for a total of 87±10 stimulations per fly.

As a negative control, we performed the same type of experiment in flies lacking the PEN1 Gal4 line, and we confirmed that the ATP puff only rarely preceded bump jumps in this genotype (*Figure 4B*). This negative result demonstrates that the bump jump in the experimental genotype is primarily driven by P2X2 expression in PEN1 neurons. Bump jumps in control flies are likely coincidental, as the bump is often moving in a typical EPG imaging experiment. In control flies, in the few trials where the bump jumped, the bump returned to its initial location 35% of the time. These returns are not unexpected, because the bump's location is thought to be continuously compared to an internal goal location stored downstream from EPG neurons, and any deviation from the fly's angular goal is corrected via compensatory turning maneuvers.

Note that, in these experiments, EPG neurons expressed GCaMP6f (under the control of *60D05-LexA*), PEN1 neurons expressed $P2X_2$ and HaloTag (under the control of *VT032906-Gal4*), and DNa02 neurons also expressed $P2X_2$ and HaloTag (under the control of *SS00730*). We took several steps to verify that ATP directly stimulated PEN1 neurons but not DNa02 neurons. First, we used dye to check that the ATP bolus was confined to the protocerebral bridge. Second, we confirmed that DNa02 neurons were not depolarized during the period when PEN1 neurons were depolarized and the bump, therefore, jumped, i.e., in the first 0.5 s after ATP ejection (mean DNa02 voltage change in the 0.5 s after ATP ejection was 0.12±0.27 mV in experimental flies and 0.14±0.27 mV in genetic controls lacking the PEN1 Gal4 driver). Rather, DNa02 neurons were only excited just before the fly made a compensatory steering maneuver, which usually occurred 1.0–1.5 s after the bump jump. Third, we verified that DNa02 was only recruited when the bump returned to its initial location via a clockwise path; if ATP had been directly exciting DNa02, then the side where DNa02 was active should not be contingent on the bump's path.

## Fictive odor and fictive heat stimuli

In *Figure 5* (and *Figure 5—figure supplements 1 and 2*), we used the LexA system to express *LexAOp-CsChrimson::mVenus* under the control of either *Orco-LexA* (*Lai and Lee, 2006*) or *Gr28b.d--LexA* (*Frank et al., 2015*). To illuminate the antennae, we fixed a pair of 50 µm fiber optic cannulas (10 mm long, FG050UGA, Thorlabs) to the underside of the recording platform, so that each fiber was directed at one antenna. Alignment of each cannula was confirmed in every experiment by looking for uniform and symmetric illumination of each antenna (*Gaudry et al., 2013*). A 660 nm light source was fiber-coupled to each cannula (M660F1, Thorlabs). Stimulation lasted for 0.5 s, with an inter-stimulus interval of 11 s.

## Stochastic ReaChR expression

We expressed ReaChR stochastically in DNa02 neurons to obtain unilateral optogenetic activation of these neurons. Flies expressed FLP under the control of a heat-shock promoter, and they expressed *UAS-FRT.mCherry.FRT.ReachR::citrine* under the control of the DNa02 split-Gal4 line (*SS00730*). These flies were allowed to develop at room temperature (~21 °C), as we found that this was sufficient to produce stochastic labeling. We took 1-d-old female virgin flies from this culture, and we fixed them into our standard platform, but no dissection was performed prior to making behavioral measurements, i.e., the head cuticle remained intact. After the flies acclimated to the spherical treadmill,

orange light (617 nm, M617L3, Thorlabs) was delivered from above the head, so that the left and right sides of the head were illuminated symmetrically. The power density at the fly's head was 68 μW/mm². Light was switched on for 0.5 s every 11 s for a total of 150 trials. The experimenter remained blind to the fly's behavior throughout these trials. At the end of these trials, the dorsal cuticle was dissected away, and ReaChR expression was scored. In a total of 73 experiments, we observed bilateral expression in 47 brains, left-only expression in nine brains, right-only expression in eight brains, and no expression in eight brains. After ReaChR expression was scored, flies were divided into data analysis categories and behavioral data was analyzed accordingly.

## EM reconstruction

To identify DNa02 in the (FAFB) dataset (*Dorkenwald et al., 2023*; *Dorkenwald et al., 2022*; *Scheffer et al., 2020*; *Schlegel et al., 2023*; *Zheng et al., 2018*), we traced all the neurites with large cross-sections in the tract where DNa02's primary neurite resides, in the vicinity of the ventral lateral accessory lobe. All these primary neurites were reviewed by two experts, who agreed that only one neuron resembled DNa02. We then fully reconstructed the skeleton of this DNa02-like neuron. Next, we confirmed that this neuron bore an excellent resemblance to skeletons we traced from two 3D confocal images of the DNa02-specific split-Gal4 line (*SS00730*) driving CD8::GFP expression. The first of these images was provided to us by Gwyneth Card (*Namiki et al., 2018*). The second image we obtained via immunostaining and confocal imaging using standard procedures. We registered both 3D images to a common template brain, JFRC2013 (*Bates et al., 2020*), using the neuropil counterstain (anti-Bruchpilot immunostaining). Image registration was carried out as described previously (*Cachero et al., 2010*) using the CMTK registration suite (https://www.nitrc.org/projects/cmtk). Single neurons were manually traced from 3D confocal images using the Simple Neurite Tracer plugin (*Longair et al., 2011*) in ImageJ. For comparison with light-level data, the EM skeleton was also registered onto the JFRC2013 template brain. DNa01 was identified using similar approaches, which are now detailed in *Stürner et al., 2024*; note that this pair of neurons was identified as present in the line *SS00731* and distinct from the neurons (incorrectly) labeled as DNa01 in hemibrain:v1.2.1 dataset. In FAFB/FlyWire v783, root IDs of these cells are as follows: DNa02R 720575940604737708, DNa02L 720575940629327659, DNa01R 720575940644438551, and DNa01L 720575940627787609. DNa01 and DNa02 outputs in the ventral nerve cord were retrieved from the male VNC connectome (manc v1.2.1) using neuprint+ (*Plaza et al., 2022*) and analyzed using neuprint-python (https://connectome-neuprint.github.io/neuprint-python), *Berg and Schlegel, 2017*. DNa01 and DNa02 inputs in the brain were retrieved from the female brain connectome (FAFB/FlyWire), specifically materialization version 783; cell and synapse counts were analyzed using fafbseg (*Bates et al., 2020*). The major central complex and olfactory/thermosensory pathways upstream from DNa02 were analyzed using Codex and fafbseg. Neurotransmitters associated with FAFB cells are taken from automatic predictions (*Eckstein et al., 2024*). To select cell types for inclusion in *Figure 7C*, we identified all individual cells postsynaptic to PFL3 and presynaptic to DNa02, discarding any unitary connections with <5 synapses. We then grouped unitary connections by cell type, and then summed all synapse numbers within each connection group (e.g. summing all synapses in all PFL3→LAL126 connections). We then discarded connection groups having <200 synapses or <1% of a cell type's pre- or postsynaptic total. Reported connection weights are per hemisphere, i.e., half of the total within each connection group. For *Figure 7F*, we did the same, but now discarding connection groups having <70 synapses or <0.4% of a cell type's pre- or postsynaptic total. In *Figure 7—figure supplement 1*, we used the same procedures for analyzing connections onto DNa01.

## Immunofluorescence confocal microscopy

The brain-and-VNC was dissected in external saline, fixed in 4% paraformaldehyde (Electron Microscopy Sciences) in PBS (Thermo Fisher Scientific) for 15 min at room temperature, and washed with PBS. After a 20 min block in 5% normal goat serum (Sigma-Aldrich) in PBS with 0.44% Triton-X (Sigma-Aldrich), samples were incubated for ~24 hr with primary antibodies in blocking solution, washed with PBS, and incubated for ~24 hr with secondary antibodies. Primary antibodies were chicken anti-GFP (1:1000, Abcam #13970, RRID:AB_300798) and mouse anti-Bruchpilot (1:30, nc82, DSHB #nc82, RRID:AB_2392664). Secondary antibodies were Alexa Fluor 488 anti-chicken and Alexa Fluor 633 anti-mouse (1:250, Invitrogen #11039 and #21050, RRID:AB_2534096 and RRID:AB_2535718). Finally,

samples were rinsed, mounted in antifade medium (Vectashield, Vector Laboratories), and imaged on a Leica confocal microscope (SP8X) with a 20x or 40x oil-immersion objective.

## Data processing and analysis

### Data alignment

Behavioral, stimulus, and imaging data were aligned using triggers acquired on the same NI-DAQ board.

### Data analysis – spherical treadmill data preprocessing

Kinematic data used in computing linear filters, signal autocorrelations, LOWESS-fit scatter-plots, and event-triggered analysis were processed as follows. First, the zero-point of each kinematic signal was corrected for small artifactual offsets by finding the median of periods of inactivity, defined as regions with an instantaneous difference of less than 0.025°/s. The offset value during these periods was largely consistent throughout the entire experiment, so a global subtraction was used to remove this offset for each experiment.

We then lightly smoothed each kinematic variable by convolving with a 50 ms Gaussian kernel bounded from $-3.5\,\sigma$ to $+3.5\,\sigma$, and re-normalized. In order to remove remaining high frequency noise while preserving large signal excursions, we then used Gaussian process smoothing. This smoothed the data by fitting the time series to a Gaussian random walk process and taking the maximum a posteriori probability (MAP) estimate for each time point, as found with the L-BFGS-B optimizer. The model was implemented with PyMC3 (*Salvatier et al., 2016*). Briefly, the model is as follows:

$$z_i \sim Normal\left(z_{i-1} + \mu,\ (1-\alpha)*\sigma^2\right) \tag{1}$$

$$y_i \sim Normal\left(z_i,\ \alpha*\sigma^2\right) \tag{2}$$

Where each time point $y_i$ was modeled as a draw from a normal distribution parameterized with mean $z_i + \mu$ and standard deviation $\alpha*\sigma^2$, where $\alpha$ acts as a factor to assign signal variance to a moving average or normally distributed noise. All variables except $\alpha$ were fit using the approximate inference software; μ was given a large initial prior standard deviation (100,000, or 5–6 times the largest value present in the data) and it quickly converged to a stable value. We set $\alpha$ to 0.2 to assign 20% of signal variance to noise and the remaining 80% to the random walk. All time series were scrutinized to ensure proper baseline correction and smoothing.

Total speed was calculated by taking the absolute value of rotational velocity ($v_r$) sideways velocity ($v_s$), and forward velocity ($v_f$), and then summing these values:

$$\text{total speed} = |v_r| + |v_s| + |v_f| \tag{3}$$

where $v_r$, $v_s$, and $v_f$ are all expressed in units of °/s (i.e. sideways and forward velocity were not yet converted into units of mm/s). This metric quantifies the overall level of fly movement, irrespective of the direction of movement.

### Data analysis - electrophysiology data preprocessing

To compute mean changes in membrane voltage relative to baseline (mean Δvoltage) in *Figure 2E* (and *Figure 1—figure supplement 1*, *Figure 3—figure supplement 2*), we first removed spikes by median-filtering the raw voltage trace. To compute firing rates in *Figures 2C-E and 3B-C*, *Figures 4–6* (and *Figure 3—figure supplement 1*), we first detected spikes by using a relative prominence metric in the Matlab findpeaks function. We then counted spikes in 10 ms non-overlapping bins and smoothed them with an exponential filter using the smoothts function in Matlab with a 30 ms window size to generate a continuous waveform of firing rate versus time.

For *Figures 1, 2B, 3D and E* (and *Figure 1—figure supplement 2*), firing rate was calculated as follows. First, data were conservatively band-pass zero-phase filtered using a first-order Butterworth filter with natural (3 dB) frequency of 100 Hz. Filtered time series were then normalized to a 500 ms rolling estimate of the median absolute deviation (MAD). MAD is defined as the median of absolute deviations from the median of the entire dataset. We used a corrected version of this estimate to maintain asymptotically normal consistency by multiplying by 1.4826. This aids spike detection when

firing rates are high. Spikes were then detected using the relative prominence metric in the find_peaks function in the SciPy Python library, where the prominence of a given spike was defined over a 10 s period. Prominence values were selected for each experiment and neuron. While requiring threshold tuning for some experiments, this method was robust in all DNa01 and DNa02 recordings. To estimate the firing rate, detected spikes were then binned at 1.25 ms and smoothed by convolving with a 2.5 ms Gaussian kernel as described above. All steps of firing rate calculation were scrutinized to ensure proper behavior by plotting periods known to be difficult or easy to estimate (i.e. small inter-spike interval or high MAD).

To compute autocorrelations (*Figure 1—figure supplement 2*) and linear filters (*Figures 1, 2B, 3D and E*, and *Figure 1—figure supplement 2*), we processed electrophysiology data as follows. We first estimate an initial 'offset' by taking the median of the first 30–60 s. We then median-filtered the entire time series using a kernel length of 35 ms. This window length was empirically selected to optimally preserve low-frequency voltage modulations while excluding most of the spike waveform. We then removed high frequency artifacts remaining after median filtering by convolving with a 5 ms Gaussian kernel (bounded from $-3.5\sigma$ to $+3.5\sigma$ and re-normalized). The resulting time series was then detrended by subtracting the average of linear fits to voltage data over every 30 s and 120 s, and then adding back the initial offset. All time series were scrutinized to ensure that median filtering and detrending were successful.

To compute neuron-behavior relationships, firing rates and membrane voltage values were downsampled to the same sampling rate as the spherical treadmill data. This was done by applying an upsampling by a factor of 4, applying a linear-phase finite impulse response filter, and then downsampling.

## Data analysis - signal autocorrelation analysis
Kinematic variable autocorrelation and neuron firing rate autocorrelation were calculated using the correlate function in the python library SciPy (*Figure 1—figure supplement 2*, https://scipy.org/) or using the xcorr function in Matlab (*Figure 2A*).

## Data analysis - linear filter analysis
Linear filters (*Figures 1, 2B, 3D and E*, and *Figure 1—figure supplement 2*) were calculated using the sampling rate of the spherical treadmill data, meaning that electrophysiology data were downsampled as described in the preprocessing section. We calculated these linear filters by treating one signal as input and the other as output. We found the Fast Fourier Transform (FFT) of 1-sample overlapping 4 s windows of both input and output, and then computed.

$$F\left(\Omega\right) = \left(input^{*}\left(\Omega\right) \times output\left(\Omega\right)\right) / \left(input^{*}\left(\Omega\right) \times input\left(\Omega\right)\right) \tag{4}$$

where $input^{*}$ represents the complex conjugate of $input$. The denominator of this equation is the input power spectrum, and the numerator is the cross-correlation between input and output in the frequency domain. We then applied the inverse of the Fast Fourier Transform to $F$ to yield the linear filter relationship between input and output. For filter analyses, all behavioral variables were represented in units of °/s (i.e. sideways velocity and forward velocity were not converted from °/s to mm/s). This was done so that all filters had the same units (°/s in *Figure 1* and *Figure 1—figure supplement 2B–F*, or s/° in *Figure 1—figure supplement 2H–L*), allowing direct comparison of different filter amplitudes. Because walking behavior appeared consistent over the course of the experiments in general, linear filters were found using the first 20% of the experiment, further restricting this data to exclude times when there was optogenetic stimulation.

Our input and output signals have little high-frequency content, and so division in the frequency domain introduces some noise into the linear filter. Moreover, each class of input-output relationship has its own characteristic frequency content. For this reason, we low-pass filtered the linear filters to remove this noise differently for each class. Behavior→Neuron filters were filtered with a 0[th] order Slepian window of 6 Hz bandwidth, whereas Neuron→Behavior filters were low pass filtered using a 0[th] order Slepian window of 15 Hz. These windows maximize energy in the central lobe and were empirically selected to avoid distortion of spectral content within the window.

To generate behavioral predictions, we convolved each cell's firing rate estimate with its Neuron→Behavior filters using the final 80% of the experiment, again excluding periods with optogenetic

stimulation. To find the variance explained by each filter, we linearly regressed the predicted output against the actual output and took the $R^2$ value (*Figure 1H*, *Figure 1—figure supplement 2F and L*). Since each experiment differed in signal-to-noise, and low pass filtering was applied globally, the length of filter used in prediction was optimized within valid bounds for each neuron (100ms to 4 s). In general, we found that a shorter filter was optimal for rotational velocity and sideways velocity, whereas a longer filter was optimal for forward velocity and total speed, consistent with the differing timescales of the autocorrelograms of these variables (*Figure 1—figure supplement 2*). In scatter plots of predicted turning, we excluded periods of inactivity for clarity (*Figure 3E*). We defined inactivity here using a histogram-based method to robustly classify movement versus non-movement. Briefly, we constructed a histogram using the outlier-robust Freedman Diaconis estimator to generate bins centered around the modal timeseries value for each of the three kinematic variables, then found all points in the fly's behavior where all three variables were within these bin edges (close to zero net ball motion). For generating a two-cell behavioral prediction from DNa02 dual recordings, we simply summed the predictions of both filters, again using only periods when the fly was active (*Figure 3E*).

## Data analysis - transitions between movement and immobility

To generate *Figure 6B*, we defined total speed (in units of °/s) according to *Equation 3*. We applied a threshold of 75°/s to the fly's total speed in each experiment in the increasing direction (to detect 'starts') and also the decreasing direction (to detect 'stops'). We then discarded starts where the fly was not consistently stopped before starting (i.e. in the window 750 ms before threshold crossing, 90% of data points were not less than or equal to ½ the threshold value), and we also discarded starts where the fly was not consistently moving after threshold crossing (i.e. in the window 750 ms after threshold crossing, 90% of the data points were not greater than the threshold value). Conversely, we also discarded stops where the fly was not moving consistently before stopping (i.e. in the window 750 ms before threshold crossing, 90% of the data points were not above the threshold value), and we also discarded stops where the fly was not consistently stopped after the threshold crossing (i.e. in the window after threshold crossing, 90% of the data points were not less than or equal to ½ the threshold value). Our conclusions based on this analysis were generally insensitive to the length of the window chosen. Finally, we aligned events by the time of threshold crossing, and we averaged data across events within each experiment before averaging across experiments.

## Data analysis – colormaps of behavioral data in 2-neuron space

To generate the colormaps in *Figures 2 and 3* (and *Figure 3—figure supplement 1*), we shifted the neural data forward by 150 ms to account for the delay in behavior relative to neural data. We then divided both neural data and behavioral data into 50 ms non-overlapping time windows. We computed the average firing rate or membrane voltage in each window; these values were then used to bin the behavioral data in two-dimensional firing rate space or two-dimensional membrane voltage space.

## Data analysis - central complex stimulation

For the imaging data in *Figure 4*, imaging planes through the ellipsoid body (EB) were resliced to obtain coronal sections. Using custom Matlab code, we manually divided the EB into eight equal wedge-shaped sectors. We then calculated ΔF/F for each EB sector, defining F as the average over time of the lower half of the raw fluorescence values for that sector.

We estimated the position of the bump of activity in EPG neurons by computing a population vector average (PVA) (*Seelig and Jayaraman, 2015*). Our experimental design required that we focus on trials where the bump was clearly visible and stable, jumped after the ATP puff, and then returned to its initial position; our goal was to study DNa02 neuron responses during this sequence of events. We, therefore, discarded trials where the bump amplitude faded during the trial – specifically, where PVA magnitude dropped below the seventh percentile (for that experiment) for more than 1 s during the trial. We then discarded trials where the bump was already moving during the 1 s prior to the ATP puff, meaning that the standard deviation of the bump's position during that period was >1.5 sectors. At this point, we computed the 'initial position' of the bump in each trial as its mean position during the 1 s prior to the ATP puff. Next, we discarded trials where the bump did not jump in response to the ATP puff, i.e., where the bump's position did not move by at least 0.5 sectors. We also discarded

trials where the bump did not return to dwell within 0.5 sectors of its initial position (i.e. its position before the bump jump) for at least 0.5 s. Our results were not especially sensitive to the precise values of any of these data inclusion criteria.

We defined the bump's return period as the time window beginning with the maximum excursion of the bump (from its initial position) and ending with the bump's return to a position within 0.5 sectors of its initial position. We found the time point during each trial where the bump's return speed was maximal. We then aligned trials by this time point before averaging data across trials and experiments. We confirmed that counterclockwise bump movements are generally associated with rightward behavioral turns, while clockwise bump movements are associated with leftward behavioral turns, as described previously (*Turner-Evans et al., 2017*).

## Data analysis - optogenetic inactivation of DNa01 or DNa02

We removed 27 s video chunks where the fly moved less than one body length. We then cropped each chunk, with the field of view centered around the center of the fly, and we used these videos as inputs to the DeepLabCut (*Mathis et al., 2018*) annotation software. In 1300 frames from 12 flies, we manually annotated eight body parts in each frame: the tarsi of each leg, the posterior tip of the abdomen, and the center of the anterior edge of the head. Frames were chosen for manual annotation to maximize the diversity of walking angles in the field of view. Training was performed on 95% of this 1300-frame dataset, and the remaining 5% was used to visually evaluate model performance by comparing the locations of manually labeled points and auto-labeled points. When the model mislabeled frames, it was generally because of an unusual walking angle or lighting condition, and so we added frames with these conditions to our training set and recomputed the detection model to improve performance in these situations. We also excluded stretches of data when the fly was grooming, jumping, or walking on the ceiling of the arena, because all these events reduced the model's performance. We labeled a leg 'in stance' if the smoothed instantaneous velocity of the limb was <8 mm/s (averaged over three consecutive frames); otherwise, the leg was labeled 'in swing'.

### Sideways speed and rotational speed (Figure 8E)

Taking data from all three genotypes, we divided each video chunk into non-overlapping 500 ms windows. We then computed the vector between the head and the abdomen tip. We used this vector to compute the fly's forward velocity, sideways speed, and rotational speed, for each frame, averaged over all frames in the window. We discarded windows where the fly's average forward velocity was <2.0 mm/s. To determine whether there was a significant effect of DNa01 silencing on sideways speed or rotational speed, we performed a two-factor ANOVA with genotypes (DNa01/control) and light (on/off) as factors, and we examined the p-values associated with genotype ×light interaction, after p-values were corrected for multiple comparisons (Bonferroni-Holm correction, m=2 tests). We used the same procedure to determine whether there was a significant effect of DNa02 silencing.

### Leg movements associated with body rotation events (Figure 8F, G, Figure 8—figure supplement 1)

We analyzed five metrics intended to capture the multi-leg kinematic features associated with body rotations. These metrics were based on previous descriptions of leg movements during body rotations (*DeAngelis et al., 2019*; *Strauss and Heisenberg, 1990*). Importantly, we determined the number of metrics we would test, and we fixed the exact definitions of those metrics, based on data from control genotypes alone. Our goal was to identify leg-kinematic features associated with body rotation in control flies, in order to subsequently determine if any of these features were altered when DNs were silenced. To analyze leg-kinematic features associated with body rotation, our first step was to detect body-rotation events by searching forward in time for moments where the fly's rotational speed exceeded a threshold of 20°/s and stayed above that threshold for at least 0.1 s. We then extracted a 500 ms window of data starting 100ms prior to the threshold crossing (We explored several window sizes and chose 500 ms because it produced the strongest relationship between the multi-leg metrics and rotational movement, considering all five metrics taken together.). Next, for each leg, we detected every complete swing epoch and every complete stance epoch in the window. Then, in each window, we computed the following metrics (iF = inner front, iM = inner middle, iB = inner back, oF = outer front, oM = outer middle, oB = outer back):

1. *Stance direction* (***Figure 8—figure supplement 1***, row 1): The vector pointing backward along the body's long axis was taken as 0°. We measured each leg's movement direction (in body-centric coordinates) during each stance. When the fly is walking forward, leg stance directions are close to 0° (i.e., every leg is moving almost straight in the backward direction relative to the body). During left turns, the legs in stance tend to move rightward (0° < $\theta$ < 180°), while conversely, during right turns, the legs in stance tend to move leftward (0° > $\theta$ > −180°); see ***Figure 8—figure supplement 1B1***. The stance direction metric was defined as the mean of the stance directions of the oF, iF, and iM legs (averaged over all epochs in the window).

2. *Swing direction* (***Figure 8—figure supplement 1***, row 2) The vector pointing forward along the body's long axis was taken as 0°. We measured each leg's movement direction (in body-centric coordinates) during each swing. When the fly is walking forward, leg swing directions are close to 0° (i.e. every leg is moving almost straight in the forward direction relative to the body). During left turns, the legs tend to swing leftward (0° < θ < −180°), while conversely, during right turns, the legs tend to swing rightward (0° > $\theta$ >180°); see ***Figure 8—figure supplement 1B2***. The swing direction metric was defined as the mean of the stance directions of the oF, iF, and iM legs (averaged over all epochs in the window).

3. *Swing distance* (***Figure 8—figure supplement 1***, row 3) We measured the distance each leg moved during its swing epochs. The swing distance metric was defined as the mean of the swing distances of the OF and OM legs (averaged across swing epochs), divided by the mean swing distances of the iM and iB legs (averaged across epochs in the window).

4. *Stance duration **Figure 8—figure supplement 1***, row 4 and ***Figure 8F and G*** We measured the time each leg spent in each of its stance epochs. The stance duration metric was defined as the stance duration of the iB leg (averaged over stance epochs), divided by the mean stance durations of the iF, oF, oM, and oB legs (averaged across epochs in the window).

5. *Swing duration* (***Figure 8—figure supplement 1***, row 5) We measured the time each leg spent in each of its swing epochs. The swing duration metric was defined as the mean swing duration of the iM and iB legs (averaged across epochs in the window).

To determine whether there was a significant effect of DNa01 silencing on any metric, we performed a two-factor ANOVA with genotypes (DNa01/control) and light (on/off) as factors, and we examined the p-values associated with genotype ×light interaction, after p-values were corrected for multiple comparisons (Bonferroni-Holm correction, m=5 tests). We used the same procedure to determine whether there was a significant effect of DNa02 silencing.

## Step frequency, step length, and forward velocity (Figure 8H)

We divided each video chunk into non-overlapping 500 ms windows. We discarded windows where the fly's time-averaged forward velocity was <2.0 mm/s. Step frequency was defined as the mean frequency of stance onset for all six legs, averaged across all complete stride epochs within that window, for all legs. Step length was defined as the mean stride distance of all six legs, averaged across all complete stride epochs within that window, for all legs. Forward velocity was defined as the velocity of the vector connecting the abdomen tip to the head, in the direction of that vector, for each frame, and then averaged over the window. To determine whether there was a significant effect of DNa01 silencing on any of these three variables (step frequency, step length, forward velocity), we performed a two-factor ANOVA with genotypes (DNa01/control) and light (on/off) as factors, and we examined the p-values associated with genotype ×light interaction, after p-values were corrected for multiple comparisons (Bonferroni-Holm correction, m=3 tests). We used the same procedure to determine whether there was a significant effect of DNa02 silencing.

For the leg-centric metrics above, if any leg included in a given metric did not complete at least one of the relevant epochs during a particular time window, then that window was omitted. Moreover, for the ANOVAs described above, if a fly did not contribute at least five time windows to the dataset for that particular metric, then the fly was omitted. The number of flies that contributed data to each panel was as follows (control/DNa01/DNa02 genotypes): ***Figure 8E*** (7/11/6); ***Figure 8F and G*** (5/8/6); ***Figure 8H*** – step frequency (6/8/6), step length (6/10/6), forward velocity (7/11/6). For ***Figure 8—figure supplement 1***, the values are: row 1 (6/9/6); row 2 (6/9/6); row 3 (6/9/6); row 4 (5/8/6); row 5 (6/9/6).

## Post hoc statistical tests

When any ANOVA yielded a significant genotype ×light interaction, we followed up by performing a post hoc Tukey test to determine whether, for each genotype, there was a significant effect of light on/off. Finally, to confirm that it was reasonable to treat each time window as an independent datapoint, we performed a separate two-factor ANOVA with fly ID and light as factors, and we verified that the variance due to fly ID is much smaller than total variance.

## Data inclusion

In the analysis of electrophysiology experiments, we excluded cells where the recording lasted ≤ 15 min, to ensure that our measurements from each cell were based on adequate sampling. Late in a recording, the membrane voltage sometimes became depolarized, which we interpret as a sign of poor recording quality; we, therefore, discarded any extended epoch where the membrane voltage was more depolarized than –33 mV.

In *Figure 5* (and *Figure 5—figure supplement 1*, and *Figure 5—figure supplement 2*), flies were excluded that displayed highly asymmetric stimulus-evoked steering behavior. For example, we excluded a fly if, on average, she displayed significant stimulation-evoked turns to the right but not to the left. These cases are likely due to the asymmetric positioning of the fiber optic filaments near the antennae. This excluded 2/6 flies for fictive odor experiments and 3/7 flies for fictive warming experiments.

In *Figure 4*, flies were excluded if there were <6 trials that passed the checks described above (see *Data analysis – central complex stimulation*), to ensure that our measurements from each cell were based on adequate sampling. This excluded 4 of 8 recordings.

In single-cell recordings (*Figures 1 and 4–6*; *Figure 1—figure supplement 2*, *Figure 5—figure supplement 1*, and *Figure 5—figure supplement 2*), we excluded one DNa01 cell (out of 8 total) and one DNa02 cell (out of 11 total) where the recording quality fluctuated substantially over time.

These inclusion criteria were not pre-determined.

## Acknowledgements

We are grateful to Gwyneth Card, Michael Dickinson, Hiro Namiki, and Wyatt Korff for sharing descending neuron morphology data and split-Gal4 drivers (*SS00730* and *SS00731*) pre-publication. Isabel D'Alessandro assisted with the process of using these drivers to identify DNa02 in the FAFB data set. We thank Katharina Eichler, Marta Costa, and Gregory Jefferis for sharing the identification of DNa01 prior to publication. Luke D Lavis and Jonathan B Grimm provided SiR110-HaloTag dye pre-publication. Julian Ng, Sebastian Cachero, and Gregory Jefferis shared *pJFRC81-td3-Halo7::CAAX(attP18)* flies pre-publication. Tzumin Lee shared *Orco-LexA* flies, Marco Gallio shared *Gr28b.d-LexA* flies, Gero Miesenböck shared *UAS-P2X$_2$* flies, Tom Clandinin shared *hsFLP.70* flies, and Mike Crickmore shared *UAS-GtACR1::eYFP(VK00005)* flies. Jonathan Green provided excellent advice on central complex stimulation. We thank Anna Li, Isabel Haber, Peter Gibb, Mert Erginkaya, Saba Ali, Kelli Fairbanks, Tansy Yang, Emily Tenshaw, Markus Pleijzier, Imaan Tamimi, Eugenia Chiappe, Vivek Jayaraman, Barry Dickson, and Gwyneth Card for assistance in manual reconstruction of DNa02 inputs, prior to the advent of large-scale connectome data. Tom Kazimiers helped transfer our EM reconstructions to the FAFB-v14 community workspace. Douglas Hayden provided statistical advice. Richard Mann and members of the Wilson lab provided feedback on the manuscript. FAFB tracing environment and analysis tools were funded in part by NIH grant 1RF1MH120679 to Davi Bock and Greg Jefferis, with software development and administrative support provided by Tom Kazimiers (Kazmos GmbH) and Eric Perlman (Yikes LLC). Development of the natverse, including the fafbseg package has been supported by the NIH BRAIN Initiative (grant 1RF1MH120679-01), NSF/MRC Neuronex2 (NSF 2014862/MC_EX_MR/T046279/1), and core funding from the Medical Research Council (MC_U105188491). We thank the Princeton FlyWire team and members of the Murthy and Seung labs, as well as members of the Allen Institute for Brain Science, for development and maintenance of FlyWire (supported by BRAIN Initiative grants MH117815 and NS126935 to Murthy and Seung). We also acknowledge members of the Princeton FlyWire team, the Cambridge Connectomics and Jefferis Groups, and the FlyWire consortium for neuron proofreading and annotation. AR was supported by National Research Service Award F31DC015701. SLH was supported by a National Science Foundation

Graduate Research Fellowship and a National Research Service Award F31NS106982. This work was supported by NIH grants R01DC008174, R01NS101157, and U19NS104655. RIW is an HHMI Investigator.

## Additional information

### Competing interests

Laia Serratosa Capdevila: Affiliated with Aelysia LTD; the author has no other competing interests to declare. The other authors declare that no competing interests exist.

### Funding

| Funder | Grant reference number | Author |
|---|---|---|
| National Institute of Neurological Disorders and Stroke | R01NS101157 | Rachel Wilson |
| National Institute on Deafness and Other Communication Disorders | R01DC008174 | Rachel Wilson |
| National Institute of Neurological Disorders and Stroke | U19NS104655 | Rachel Wilson |
| National Institute of Neurological Disorders and Stroke | F31NS106982 | Stephen L Holtz |
| National Institute of Neurological Disorders and Stroke | F31DC015701 | Aleksandr Rayshubskiy |

The funders had no role in study design, data collection and interpretation, or the decision to submit the work for publication.

### Author contributions

Aleksandr Rayshubskiy, Conceptualization, Data curation, Software, Formal analysis, Investigation, Visualization, Methodology, Writing – original draft, Project administration, Writing – review and editing; Stephen L Holtz, Software, Formal analysis, Writing – review and editing; Alexander S Bates, Software, Formal analysis; Quinn X Vanderbeck, Laia Serratosa Capdevila, Victoria Rockwell, Visualization; Rachel Wilson, Conceptualization, Data curation, Formal analysis, Supervision, Funding acquisition, Validation, Investigation, Visualization, Writing – original draft, Project administration, Writing – review and editing

### Author ORCIDs

Aleksandr Rayshubskiy ⓘ https://orcid.org/0009-0009-3139-7606
Rachel Wilson ⓘ https://orcid.org/0000-0001-8573-9266

Reviewer #1 (Public review): https://doi.org/10.7554/eLife.102230.3.sa1
Reviewer #2 (Public review): https://doi.org/10.7554/eLife.102230.3.sa2
Reviewer #3 (Public review): https://doi.org/10.7554/eLife.102230.3.sa3
Author response https://doi.org/10.7554/eLife.102230.3.sa4

## Additional files

### Supplementary files

MDAR checklist

## Data availability

Data are deposited at Harvard Dataverse (https://doi.org/10.7910/DVN/0NCLP1). Analysis code is deposited at https://github.com/SashaRayshubskiy/eLife_102230_analysis_code (v1.0.1; copy archived at *Rayshubskiy, 2025*) and https://github.com/wilson-lab/rayshubskiy_elife_102230_secondary_analysis_code (v1.0.0; copy archived at *Holtz, 2025*).

The following dataset was generated:

| Author(s) | Year | Dataset title | Dataset URL | Database and Identifier |
|---|---|---|---|---|
| Rayshubskiy A, Holtz SL, Wilson RI | 2025 | Data for: Aleksandr Rayshubskiy, Stephen L Holtz, Alexander S Bates, Quinn X Vanderbeck, Laia Serratosa Capdevila, Victoria Rockwell, Rachel I Wilson, 2024. "Neural circuit mechanisms for steering control in walking Drosophila." eLife 102230v2 | https://doi.org/10.7910/DVN/0NCLP1 | Harvard Dataverse, 10.7910/DVN/0NCLP1 |

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
