## [Editor Report · eLife Assessment]

This **important** work investigates how orientation signals detected in higher brain areas may be transformed into motor responses in behaving animals. The authors characterize two types of descending neurons (DNs) that connect the brain to motor units and are involved in different aspects of turning control. They further show that orientation signals act by preferentially increasing relative stimulation onto left- or right-turn-inducing DNs. These **compelling** results, together with the independent work that they have inspired, represent significant progress in our understanding of mechanisms of animal navigation.

---

## [Referee Report · Reviewer #1 (Public review)]

Summary:

The paper addresses the knowledge gap between the representation of goal direction in the central complex and how motor systems stabilize movement toward that goal. The authors focused on two descending neurons, DNa01 and 02, and showed that they play different roles in steering the fly toward a goal. They also explored the connectome data to propose a model to explain how these DNs could mediate response to lateralized sensory inputs. They finally used lateralized optogenetic activation/inactivation experiments to test the roles of these neurons in mediating turnings in freely walking flies.

Strengths:

The experiments are well-designed and controlled. The experiment in Figure 4 is elegant, and the authors put a lot of effort into ensuring that ATP puffs do not accidentally activate the DNs. They also have explained complex experiments well. I only have minor comments for the authors.

Comments on revisions:

I am happy with the revised manuscript and authors' response to our concerns. The addition of Figure S8, makes it more transparent and the revised text is now more accessible to the non-experts.

---

## [Referee Report · Reviewer #2 (Public review)]

The data is largely electrophysiological recordings coupled with behavioral measurements (technically impressive) and some gain-of-function experiments in freely walking flies. Loss-of-function was tested but has minimal effect, which is not surprising in a system with partially redundant control mechanisms. The data is also consistent with/complementary to subsequent manuscripts (Yang 2023, Feng 2024, and Ros 2024) showing additional descending neurons with contributions to steering in walking and flying.

The experiments are well executed, the results interesting, and the description clear. Some hypotheses based on connectome anatomy are tested: the insights on the pre-synaptic side - how sensory and central complex heading circuits converge onto these DNs is stronger than the suggestions about biomechanical mechanisms for how turning happens on the motor side.

Of particular interest is the idea that different sensory cues can converge on a common motor program. The turn-toward or turn-away mechanism is initiated by valence rather than whether the stimulus was odor or temperature or memory of heading. The idea that animals chose a direction based on external sensory information and then maintain that direction as a heading through a more internal, goal-based memory mechanism, is interesting but it is hard to separate conclusively.

The "see-saw", where left-right symmetry is broken to allow a turn, presumably by excitation on one side and inhibition of the other leg motor modules, is interesting but not well explained here. How hyperpolarization affects motor outputs is not clear.

The statement near Figure 5B that "DNa02 activity was higher on the side ipsilateral to the attractive stimulus, but contralateral to the aversive stimulus" is really important - and only possible to see because of the dual recordings.

Comments on revisions:

I am happy that the revised manuscript addresses all reviewers' concerns.

---

## [Referee Report · Reviewer #3 (Public review)]

Summary:

Rayshubskiy et al. performed whole-cell recordings from descending neurons (DNs) of fruit-flies to characterize their role in steering. Two DNs implicated in "walking control" and "steering control" by previous studies (Namiki et al., 2018, Cande et al., 2018, Chen et al., 2018) were chosen by the authors for further characterization. In-vivo whole-cell recordings from DNa01 and DNa02 showed that their activity predicts spontaneous ipsilateral turning events. The recordings also showed that while DNa02 predicts transient turns DNa01 predicts slow sustained turns. However, optogenetic activation or inactivation showed relatively subtle phenotypes for both neurons (consistent with data in other recent preprints, Yang et al 2023 and Feng et al 2024). The authors also further characterized DNa02 with respect to its inputs and show functional connection with olfactory and thermosensory inputs as well as with the head-direction system. DNa01 is not characterized to this extent.

Strengths:

(1). In-vivo recordings and especially dual recordings are extremely challenging in *Drosophila* and provide a much higher resolution DN characterization than other recent studies which have relied on behavior or calcium imaging. Especially impressive are the simultaneous recordings from bilateral DNs (Fig. 3). These bilateral recordings show clearly that DNa02 cells not only fire more during ipsilateral turning events but that they get inhibited during contralateral turns. In-line with this observation, the difference between left and right DNa02 neuronal activity is a much better predictor of turning events compared to individual DNa02 activity.

(2). Another technical feat in this work is driving local excitation in the head-direction neuronal ensemble (PEN-1 neurons), while simultaneously imaging its activity and performing whole-cell recordings from DNa02 (Fig. 4). This impressive approach provided a way to causally relate changes in the head-direction system to DNa02 activity. Indeed, DNa02 activity could predict the rate at which an artificially triggered bump in the PEN-1 ring-attractor returns to its previous stable point.

(3). The authors also support the above observations with connectomics analysis and provide circuit motifs that can explain how head direction system (as well as external olfactory/thermal stimuli) communicated with DNa02. All these results unequivocally put DNa02 as an essential DN in steering control, both during exploratory navigation as well as stimulus directed turns.

Weaknesses:

While this study makes a compelling case for the importance of DNa02 in steering control, the role of DNa01 on the other hand seems unclear based on physiology, optogenetics perturbations as well as connectome analysis. DNa01 still remains a bit mysterious regarding both its role in controlling steering maneuvers as well as what in behavioral context it would be relevant.

---

## [Author Response]

The following is the authors’ response to the original reviews

**Reviewer #1 (Public review):**
Summary:The paper addresses the knowledge gap between the representation of goal direction in the central complex and how motor systems stabilize movement toward that goal. The authors focused on two descending neurons, DNa01 and 02, and showed that they play different roles in steering the fly toward a goal. They also explored the connectome data to propose a model to explain how these DNs could mediate response to lateralized sensory inputs. They finally used lateralized optogenetic activation/inactivation experiments to test the roles of these neurons in mediating turnings in freely walking flies.Strengths:The experiments are well-designed and controlled. The experiment in Figure 4 is elegant, and the authors put a lot of effort into ensuring that ATP puffs do not accidentally activate the DNs. They also have explained complex experiments well. I only have minor comments for the authors.

We are grateful for this positive feedback.

Weaknesses:(1) I do not fully understand how the authors extracted the correlation functions from the population data in Figure 1. Since the ipsilateral DNs are anti-correlated with the contralateral ones, I expected that the average will drop to zero when they are pooled together (e.g., 1E-G). Of course, this will not be the case if all the data in Figure 1 are collected from the same brain hemisphere. It would be helpful if the authors could explain this.

We regret that this information was not easy to find in our initial submission. As noted in the Figure 1D legend, Here and elsewhere, ipsi and contra are defined relative to the recorded DN(s). We have now added a sentence to the Results (right after we introduce Figure 1D) that also makes this point.

(2) What constitutes the goal directions in Figures 1-3 and 8, as the authors could not use EPG activity as a proxy for goal directions? If these experiments were done in the dark, without landmarks, one would expect the fly's heading to drift randomly at times, and they would not engage the DNa01/02 for turning. Do the walking trajectories in these experiments qualify as menotactic bouts?

Published work (Green et al., 2019) has shown that, even in the dark, flies will often walk for extended periods while holding the bump of EPG activity at a fixed location. During these epochs, the brain is essentially estimating that the fly is walking in a straight line in a fixed direction. (The fact that the fly is actually rotating a bit on the spherical treadmill is not something the fly can know, in the dark.) Thus, epochs where the EPG bump is held fixed are treated as menotactic bouts, even in darkness.

Our results provide additional support for this interpretation. We find that, when flies are walking in darkness and holding the bump of EPG activity at a fixed location, they will make a corrective behavioral turning maneuver in response to an imposed bump-jump. This result argues that the flies are actually engaging in goal-directed straight-line walking, i.e. menotaxis, and it reproduces the findings of Green et al. (2019).

To clarify this point, we have adjusted the wording of the Results pertaining to Figure 4.

(3) In Figure 2B, the authors mentioned that DNa02 overpredicts and 01 underpredicts rapid turning and provided single examples. It would be nice to see more population-level quantification to support this claim.

In this revision, we have reorganized Figures 1 and 2 (and associated text) to improve clarity. As part of this reorganization, we have removed this passage from the text, as it was a minor point in any event.

**Reviewer #2 (Public review):**
The data is largely electrophysiological recordings coupled with behavioral measurements (technically impressive) and some gain-of-function experiments in freely walking flies. Loss-of-function was tested but had minimal effect, which is not surprising in a system with partially redundant control mechanisms. The data is also consistent with/complementary to subsequent manuscripts (Yang 2023, Feng 2024, and Ros 2024) showing additional descending neurons with contributions to steering in walking and flying.The experiments are well executed, the results interesting, and the description clear. Some hypotheses based on connectome anatomy are tested: the insights on the pre-synaptic side - how sensory and central complex heading circuits converge onto these DNs are stronger than the suggestions about biomechanical mechanisms for how turning happens on the motor side.Of particular interest is the idea that different sensory cues can converge on a common motor program. The turn-toward or turn-away mechanism is initiated by valence rather than whether the stimulus was odor or temperature or memory of heading. The idea that animals choose a direction based on external sensory information and then maintain that direction as a heading through a more internal, goal-based memory mechanism, is interesting but it is hard to separate conclusively.

To clarify, we mention the role of memory in connection with two places in the manuscript. First, we note that the EPG/head direction system relies on learning and memory to construct a map of directional cues in the environment. These cues are, in principle, inherently neutral, i.e. without valence. Second, we note that specific mushroom body output neurons rely on learning and memory to store the valence associated with an odor. This information is not necessarily associated with an allocentric direction: it is simply the association of odor with value. Both of these ideas are well-attested by previous work.

The reviewer may be suggesting a sequential scheme whereby the brain initializes an allocentric goal direction based on valence, and then maintains that goal direction in memory, based on that initialization. In other words, memory is used to associate valence with some allocentric direction. This seems plausible, but it is not a claim we make in our manuscript.

The "see-saw", where left-right symmetry is broken to allow a turn, presumably by excitation on one side and inhibition of the other leg motor modules, is interesting but not well explained here. How hyperpolarization affects motor outputs is not clear.

We have added several sentences to the Discussion to clarify this point. According to this see-saw model, steering can emerge from right/left asymmetries in excitation, or inhibition, or both. It may be nonintuitive to think that inhibitory input to a DN can produce an action. However, this becomes more plausible given our finding that DNa02 has a relatively high basal firing rate (Fig. 1D), and DNa02 hyperpolarization is associated with contraversive turning (Fig. 5A). It is also relevant to note that there are many inhibitory cell types that form strong unilateral connections onto DNa02 (e.g., AOTU019).

The statement near Figure 5B that "DNa02 activity was higher on the side ipsilateral to the attractive stimulus, but contralateral to the aversive stimulus" is really important - and only possible to see because of the dual recordings.

We thank the reviewer for this positive feedback.

**Reviewer #3 (Public review):**
Summary:Rayshubskiy et al. performed whole-cell recordings from descending neurons (DNs) of fruit flies to characterize their role in steering. Two DNs implicated in "walking control" and "steering control" by previous studies (Namiki et al., 2018, Cande et al., 2018, Chen et al., 2018) were chosen by the authors for further characterization. In-vivo whole-cell recordings from DNa01 and DNa02 showed that their activity predicts spontaneous ipsilateral turning events. The recordings also showed that while DNa02 predicts transient turns DNa01 predicts slow sustained turns. However, optogenetic activation or inactivation showed relatively subtle phenotypes for both neurons (consistent with data in other recent preprints, Yang et al 2023 and Feng et al 2024). The authors also further characterized DNa02 with respect to its inputs and showed a functional connection with olfactory and thermosensory inputs as well as with the head-direction system. DNa01 is not characterized to this extent.Strengths:(1) In-vivo recordings and especially dual recordings are extremely challenging in *Drosophila* and provide a much higher resolution DN characterization than other recent studies that have relied on behavior or calcium imaging. Especially impressive are the simultaneous recordings from bilateral DNs (Figure 3). These bilateral recordings show clearly that DNa02 cells not only fire more during ipsilateral turning events but that they get inhibited during contralateral turns. In line with this observation, the difference between left and right DNa02 neuronal activity is a much better predictor of turning events compared to individual DNa02 activity.(2) Another technical feat in this work is driving local excitation in the head-direction neuronal ensemble(PEN-1 neurons), while simultaneously imaging its activity and performing whole-cell recordings from DNa02(Figure 4). This impressive approach provided a way to causally relate changes in the head-direction system to DNa02 activity. Indeed, DNa02 activity could predict the rate at which an artificially triggered bump in the PEN-1 ring attractor returns to its previous stable point.(3) The authors also support the above observations with connectomics analysis and provide circuit motifs that can explain how the head direction system (as well as external olfactory/thermal stimuli) communicated with DNa02. All these results unequivocally put DNa02 as an essential DN in steering control, both during exploratory navigation as well as stimulus-directed turns.

We are grateful for this detailed positive feedback.

Weaknesses:(1) I understand that the first version of this preprint was already on biorxiv in 2020, and some of the "weaknesses" I list are likely a reflection of the fact that I'm tasked to review this manuscript in late 2024 (more than 4 years later). But given this is a 2024 updated version it suffers from laying out the results in contemporary terms. For instance, the manuscript lacks any reference to the DNp09 circuit implicated in object-directed turning and upstream to DNa02 even though the authors cite one of the papers where this was analyzed (Braun et al, 2024). More importantly, these studies (both Braun et al 2024 and Sapkal et al 2024) along with recent work from the authors' lab (Yang et al 2023) and other labs (Feng et al 2024) provide a view that the entire suite of leg kinematics changes required for turning are orchestrated by populations of heterogeneous interconnected DNs. Moreover, these studies also show that this DN-DN network has some degree of hierarchy with some DNs being upstream to other DNs. In this contemporary view of steering control, DNa02 (like DNg13 from Yang et al 2023) is a downstream DN that is recruited by hierarchically upstream DNs like DNa03, DNp09, etc. In this view, DNa02 is likely to be involved in most turning events, but by itself unable to drive all the motor outputs required for the said events. This reasoning could be used while discussing the lack of major phenotypes with DNa02 activation or inactivation observed in the current study, which is in stark contrast to strong phenotypes observed in the case of hierarchically upstream DNs like DNp09 or DNa03. In the section, "Contributions of single descending neuron types to steering behavior": the authors start off by asking if individual DNs can make measurable contributions to steering behavior. Once more, any citations to DNp09 or DNa03 - two DNs that are clearly shown to drive strong turning-on activation (Bidaye et al, 2020, Feng et al 2024) - are lacking. Besides misleading the reader, such statements also digress the results away from contemporary knowledge in the field. I appreciate that the brief discussion in the section titled "Ensemble codes for steering" tries to cover these recent updates. However, I think this would serve a better purpose in the introduction and help guide the results.

We apologize for these omissions of relevant citations, which we have now fixed. Specifically, in our revised Discussion, we now point out that:

- Braun et al. (2024) reported that bilateral optogenetic activation of either DNa02 or DNa01 can drive turning (in either direction).

- Braun et al. (2024) also identified DNb02 as a steering-related DN.

- Bidaye et al. (2020), Sapkal et al. (2024), and Braun et al. (2024) all contributed to the identification of DNp09 as a broadcaster DN with the capacity to promote ipsiversive turning.

We have also revised the beginning of the Results section titled “Contributions of single descending neuron types to steering behavior”, as suggested by the Reviewer.

Finally, we agree with the Reviewer’s overall point that steering is influenced by multiple DNs. We have not claimed that any DN is solely responsible for steering. As we note in the Discussion: “We found that optogenetically inhibiting DNa01 produced only small defects in steering, and inhibiting DNa02 did not produce statistically significant effects on steering; these results make sense if DNa02 is just one of many steering DNs.”

(2) The second major weakness is the lack of any immunohistochemistry (IHC) images quantifying the expression of the genetic tools used in these studies. Even though the main split-Gal4 tools for DNa01 and DNa02 were previously reported by Namiki et al, 2018, it is important to document the expression with the effectors used in this work and explicitly mention the expression in any ectopic neurons. Similarly, for any experiments where drivers were combined together (double recordings, functional connectivity) or modified for stochastic expression (Figure 8), IHC images are absolutely necessary. Without this evidence, it is difficult to trust many of the results (especially in the case of behavioral experiments in Figure 8). For example, the DNa01 genetic driver used by the authors is also expressed in some neurons in the nerve cord (as shown on the Flylight webpage of Janelia Research Campus). One wonders if all or part of the results described in Figure 8 are due to DNa01 manipulation or manipulation of the nerve cord neurons. The same applies for optic lobe neurons in the DNa02 driver.

This is a reasonable request. We used DN split-Gal4 lines to express three types of UAS-linked transgenes:

(1) GFP

In these flies, we know that expression in DNs is restricted to the DN types in question, based on published work (Namki et al., 2018), as well as the fact that we see one labeled DN soma per hemisphere. When we label both cells with GFP, we use the spike waveform to identify DNa02 and DNa01, as described in Figure 1—figure supplement 1.

(2) ReaChR

In these flies, expression patterns were different in different flies because ReaChR expression was stochastically sparsened using hs-FLP. Expression was validated in each fly after the experiment, as described in the Methods (“Stochastic ReaChR expression”). hs-FLP-mediated sparsening will necessarily produce stochastic patterns of expression in both DNa02 and off-target cells, and this is true of all the flies in this experiment. What makes the “unilateral” flies distinct from the “bilateral” flies is that unilateral flies express ReaChR in one copy of DNa02, whereas bilateral flies express ReaChR in both copies of DNa02. On average, off-target expression will be the same in both groups.

(3) GtACR1

In these flies, we initially assumed that GtACR1 expression was the same as GFP expression under control of the same driver. However, we agree with the reviewer’s point that these two expression patterns are not necessarily identical. Therefore, to address the reviewer’s question, we performed immunofluorescence microscopy to characterize GtACR1 patterns in the brain and VNC of both genotypes. These expression patterns are now shown in a new supplemental figure (Figure S8). This figure shows that, as it happens, expression of GtACR1 is indeed indistinguishable from the GFP expression patterns for the same lines (archived on the FlyLight website). Both DN split-Gal4 lines are largely selective for the DNs in question, with limited off-target labeling. We have now drawn attention to this off-target labeling in the last paragraph of the Results, where the GtACR1 results are discussed.

(3) The paper starts off with a comparative analysis of the roles of DNa01 and DNa02 during steering. Unfortunately, after this initial analysis, DNa01 is largely ignored for further characterization (e.g. with respect to inputs, connectomics, etc.), only to return in the final figure for behavioral characterization where DNa01 seems to have a stronger silencing phenotype compared to DNa02. I couldn't find an explanation for this imbalance in the characterization of DNa01 versus DNa02. Is this due to technical reasons? Or was it an informed decision due to some results? In addition to being a biased characterization, this also results in the manuscript lacking a coherent thread, which in turn makes it a bit inaccessible to the non-specialist.

Yes, the first portion of the manuscript focuses on DNa01 and DNa02. The latter part of the manuscript transitions to focus mainly on DNa02.

Our rationale is noted at the point in the manuscript where we make this transition, with the section titled “Steering toward internal goals”: “Having identified steering-related DNs, we proceeded to investigate the brain circuits that provide input to these DNs. Here we decided to focus on DNa02, as this cell’s activity is predictive of larger steering maneuvers.” When we say that DNa02 is predictive of larger steering maneuvers, we are referring to several specific results:

- We obtain larger filter amplitudes for DNa02 versus DNa01 (Fig. 2A-C). This means that, just after a unit change in DN firing rate, we see on average a larger change in steering velocity for DNa02 versus DNa01.

- The linear filter for DNa02 has a higher variance explained, as compared to DNa01 (Fig. 2D). This means that DNa02 is more predictive of steering.

- The relationship between firing rate and rotational velocity (150 ms later) is steeper for DNa02 than for DNa01 (Fig. 2G). This means that, if we ignore dynamics and we just regress firing rate against subsequent rotational velocity, we see a higher-gain relationship for DNa02.

Our focus on DNa02 was also driven by connectivity considerations. In the same paragraph (the first paragraph in the section titled “Steering toward internal goals”). We note that “there are strong anatomical pathways from the central complex to DNa02”; the same is not true of DNa01. This point has also been noted by other investigators (Hulse et al. 2021).

We don’t think this focus on DNa02 makes our work biased or inaccessible. Any study must balance breadth with depth. A useful general way to balance these constraints is to begin a study with a somewhat broader scope, and then narrow the study’s focus to obtain more in-depth information. Here, we began with comparative study of two cell types, and we progressed to the cell type that we found more compelling.

(4) There seems to be a discrepancy with regard to what is emphasized in the main text and what is shown in Figures S3/S4 in relation to the role of these DNs in backward walking. There are only two sentences in the main text where these figures are cited.a) "DNa01 and DNa02 firing rate increases were not consistently followed by large changes in forward velocity(Figs. 1G and S3)."b) "We found that rotational velocity was consistently related to the difference in right-left firing rates (Fig. 3B). This relationship was essentially linear through its entire dynamic range, and was consistent across paired recordings (Fig. 3C). It was also consistent during backward walking, as well as forward walking (Fig. S4)." These main text sentences imply the role of the difference between left and right DNa02 in turning. However, the actual plots in the Figures S3 and S4 and their respective legends seem to imply a role in "backward walking". For instance, see this sentence from the legend of Figure S3 "When (ΔvoltageDNa02>>ΔvoltageDNa01), the fly is typically moving backward. When (firing rateDNa02>>firing rateDNa01), the fly is also often moving backward, but forward movement is still more common overall, and so the net effect is that forward velocity is small but still positive when (firing rateDNa02>>firing rateDNa01). Note that when we condition our analysis on behavior rather than neural activity, we do see that backward walking is associated with a large firing rate differential (Fig. S4)." This sort of discrepancy in what is emphasized in the text, versus what is emphasized in the figures, ends up confusing the reader. More importantly, I do not agree with any of these conclusions regarding the implication of backward walking. Both Figures S3 and S4 are riddled with caveats, misinterpretations, and small sample sizes. As a result, I actually support the authors' decision to not infer too much from these figures in the "main text". In fact, I would recommend going one step further and removing/modifying these figures to focus on the role of "rotational velocity". Please find my concerns about these two figures below:a) In Figures S3 and S4, every heat map has a different scale for the same parameter: forward velocity. S3A is -10 to +10mm/s. S3B is -6 to +6 S4B (left) is -12 to +12 and S4B (right) is -4 to +4. Since the authors are trying to depict results based on the color-coding this is highly problematic.b) Figure S3A legend "When (ΔvoltageDNa02>>ΔvoltageDNa01), the fly is typically moving backward." There are also several instances when ΔvoltageDNa02 = ΔvoltageDNa01 and both are low (lower left quadrant) when the fly is typically moving backwards. So in my opinion, this figure in fact suggests DNa02 has no role in backward velocity control.c) Based on the example traces in S4A, every time the fly walks backwards it is also turning. Based on this it is important to show absolute rotational velocity in Figure S4C. It could be that the fly is turning around the backward peak which would change the interpretation from Figure S4C. Also, it is important to note that the backward velocities in S4A are unprecedentedly high. No previous reports show flies walking backwards at such high velocities (for example see Chen et al 2018, Nat Comm. for backward walking velocities on a similar setup).d) In my opinion, Figure S4D showing that right-left DNa02 correlates with rotational velocity, regardless of whether the fly is in a forward or backward walking state, is the only important and conclusive result in Figures S3/S4. These figures should be rearranged to only emphasize this panel.

We agree that it is difficult to interpret some of the correlations between DN activity and forward velocity, given that forward velocity and rotational velocity are themselves correlated to some degree. This is why we did not make claims based on these results in the main text. In response to these comments, we have taken the Reviewer’s suggestion to preserve Figure S4D (now Figure S3). The other components of these supplemental figures have been removed.

(5) Figure 3 shows a really nice analysis of the bilateral DNa02 recordings data. While Figure S5 [now Figure S4] shows that authors have a similar dataset for DNa01, a similar level analysis (Figures 3D, E) is not done for DNa01 data. Is there a reason why this is not done?

The reason we did not do the same analysis for DNa01 is that we only have two paired DNa01-DNa01 recordings. It turned out to be substantially more difficult to perform DNa01-DNa01 recordings, as compared to DNa02-DNa02 recordings. For this reason, we were not able to get more than two of these recordings.

(6) In Figure 4 since the authors have trials where bump-jump led to turning in the opposite direction to the DNa02 being recorded, I wonder if the authors could quantify hyperpolarization in DNa02 as is predicted from connectomics data in Figure 7.

We agree this is an interesting question. However, DNa02 firing rate and membrane potential are variable, and stimulus-evoked hyperpolarizations in these DNs tend to be relatively small (on the order of 1 mV, in the case of a contralateral fictive olfactory stimulus, Figure 5A). In the case of our fictive olfactory stimuli, we could look carefully for these hyperpolarizations because we had a very large number of trials, and we could align these trials precisely to stimulus onset. By contrast, for the bump-jump experiments, we have a more limited number of trials, and turning onset is not so tightly time-locked to the chemogenetic stimuli; for these reasons, we are hesitant to make claims about any bump-jump-related hyperpolarization in these trials.

(7) Figure 6 suggests that DNa02 contains information about latent steering drives. This is really interesting. However, in order to unequivocally claim this, a higher-resolution postural analysis might be needed. Especially given that DNa02 activation does not reliably evoke ipsilateral turning, these "latent" steering events could actually contain significant postural changes driven by DNa02 (making them "not latent"). Without this information, at least the authors need to explicitly mention this caveat.

This is a good point. We cannot exclude the possibility that DNa02 is driving postural changes when the fly is stopped, and these postural changes are so small we cannot detect them. In this case, however, there would still be an interesting mismatch between the stimulus-evoked change in DNa02 firing rate (which is large) and the stimulus-evoked postural response (which would be very small). We have added language to the relevant Results section in order to make this explicit.

(8) Figure 7 would really benefit from connectome data with synapse numbers (or weighted arrows) and a corresponding analysis of DNa01.

In response to this comment, we have added synapses number information (represented by weighted arrows) to Figures 7C, E, and F. We also added information to the Methods to explain how cells were chosen for inclusion in this diagram. (In brief: we thresholded these connections so as to discard connections with small numbers of synapses.)

We did perform an analogous connectome circuit analysis for DNa01, but if we use the same thresholds as we do for DNa02, we obtain a much sparser connectivity graph. We now show this in a new supplemental figure (Figure S9). MBON32 makes no monosynaptic connections onto DNa01, and it only forms one disynaptic connection, via LAL018, which is relatively weak. PFL3 and PFL2 make no mono- or disynaptic connections onto DNa01 comparable in strength to what we find for DNa02.

The sparser connectivity graph for DNa01 is partly due to the fact that fewer cell types converge onto DNa01 as compared to DNa02 (110 cell types, versus 287 cell types). Also, it seems that DNa01 is simply less closely connected to the central complex and mushroom body, as compared to DNa02.

(9) In Figure 8E, the most obvious neuronal silencing phenotype is decreased sideways velocity in the case of DNa01 optogenetic silencing. In Figure S2, the inverse filter for sideways velocity for DNa01 had a higher amplitude than the rotational velocity filter. Taken together, does this point at some role for DNa01 in sideways velocity specifically?

No. The forward filters describe the average velocity impulse response, given a brief step change in firing rate.

Figure 1 and Figure S2 show that the sideways velocity forward filter is actually smaller for DNa01 than for DNa02. This means that a brief step change in DNa01 firing rate is followed by only a very small sideways velocity response. Conversely, the reverse filters describe the average firing rate impulse response, given a brief step change in sideways velocity. Figure S2 shows that the sideways velocity reverse filter is larger for DNa01 than for DNa02, but this means that the relationship between DNa01 activity and sideways velocity is so weak that we would need to see a very large neural response in order to get a brief step change in sideways velocity. In other words, the reverse filter says that DNa01 likely has very little role in determining sideways velocity.

(10) In Figure 8G, the effect on inner hind leg stance prolongation is very weak, and given the huge sample size, hard to interpret. Also, it is not clear how this fits with the role of DNa01 in slow sustained turning based on recordings.

Yes, this effect is small in magnitude, which is not too surprising, given that many DNs seem to be involved in the control of steering in walking. To clarify the interpretation of these phenotypes, we have added a paragraph to the end of the Results:

“All these effects are weak, and so they should be interpreted with caution. Also, both DN split-Gal4 lines drive expression in a few off-target cell types, which is another reason for caution (Fig. S8). However, they suggest that both DNs can lengthen the stance phase of the ipsilateral back leg, which would cause ipsiversive turning. These results are also compatible with a scenario where both DNs decrease the step length in the ipsilateral legs, which would also cause ipsiversive turning. Step frequency does not normally change asymmetrically during turning, so the observed decrease in step frequency during optogenetic inhibition may just be a by-product of increasing step length when these DNs are inhibited.” We have also added caveats and clarifications in a new Discussion paragraph:

“Our study does not fully answer the question of how these DNs affect leg kinematics, because we were not able to simultaneously measure DN activity and leg movement. However, our optogenetic experiments suggest that both DNs can lengthen the stance phase of the ipsilateral back leg (Fig. 8G), and/or decrease the step length in the ipsilateral legs (Fig. 8H), either of which would cause ipsiversive turning. If these DNs have similar qualitative effects on leg kinematics, then why does DNa02 precede larger and more rapid steering events? This may be due to the fact that DNa02 receives stronger and more direct input from key steering circuits in the brain (Fig. S9). It may also relate to the fact that DNa02 has more direct connections onto motor neurons (Fig. 1B).”

**Recommendations for the authors:**

**Reviewer #1 (Recommendations for the authors):**
(1) I found the sign conventions for rotational velocity particularly confusing. Figure 3 represents clockwise rotations as +ve values, but Figure 4H represents anticlockwise rotations as positive values. But for EPG bumps, anticlockwise rotations are given negative values. Please make them consistent unless I am missing something obvious.

Different fields use different conventions for yaw velocity. In aeronautics, a clockwise turn is generally positive. In robotics and engineering of terrestrial vehicles, a counterclockwise turn is generally positive. Historically, most *Drosophila* studies that quantified rotational (yaw) velocity were focused on the behavior of flying flies, and these studies generally used the convention from aeronautics, where a clockwise turn is defined as a positive turn. When we began working in the field, we adopted this convention, in order to conform to previous literature. It might be argued that walking flies are more like robots than airplanes, but it seemed to us that it was confusing to have different conventions for different behaviors of the same animal. Thus, all of the published studies from our lab define clockwise rotation as having positive rotational velocity.

Figure 4 focuses on the role of the central complex in steering. As the fly turns clockwise (rightward), the bump of activity in EPG neurons normally moves counterclockwise around the ellipsoid body, as viewed from the posterior side (Turner-Evans et al., 2017). The posterior view is the conventional way to represent these dynamics, because (1) we and others typically image the brain from the posterior side, not the anterior side, and (2) in a posterior view, the animal’s left is on the left side of the image, and vice versa. We have added a sentence to the Figure 4A legend to clarify these points.

Previous work has shown that, when an experimenter artificially “jumps” the EPG bump, this causes the fly to make a compensatory turn that returns the bump to (approximately) its original location (Green et al., 2019). Our work supports this observation. Specifically, we find that clockwise bump jumps are generally followed by rightward turns (which drive the bump to return to its approximate original location via a counterclockwise path), and vice versa. This is noted in the Figure 4D legend. Note that Figure 4D plots the fly’s rotational velocity during the bump return, plotted against the initial bump jump.

Figure 4H shows that clockwise (blue) bump returns were typically preceded by leftward turning, counter-clockwise (green) bump returns were preceded by rightward turning, as expected. This is detailed in the Figure 4H legend, and it is consistent with the coordinate frame described above.

(2) It would be helpful to have images of the DNa01 and DNa02 split lines used in this paper, considering this paper would most likely be used widely to describe the functions of these neurons. Similarly, images of their reconstructions would be a useful addition.

High-quality three-dimensional confocal stacks of all the driver lines used in our study are publicly available. We have added this information to the Methods (under “Fly husbandry and genotypes”). Confocal images of the full morphologies of DNa01 and DNa02 have been previously published (Namiki et al., 2018). Figure 1A is a schematic that is intended to provide a quick visual summary of this information.

EM reconstructions of DNa01 and DNa02 are publicly accessible in a whole-brain dataset (https://codex.flywire.ai/) and a whole-VNC dataset (https://neuprint.janelia.org/). Both datasets are referenced in our study. As these datasets are easy to search and browse via user-friendly web-based tools, we expect that interested readers will have no difficulty accessing the underlying datasets directly.

**Reviewer #2 (Recommendations for the authors):**
(1) The description of the activity of the DNs that they "PREDICT steering during walking". This is an interesting word choice. Not causes, not correlates with, not encodes... does that mean the activity always precedes the action? Does that mean when you see activity, you will get behavior? This is important for assessing whether the DN activity is a cause or an effect. It is good to be cautious but it might be worth expanding on exactly what kind of connection is implied to justify the use of the word 'predict'.

Conventionally, “predict” means “to indicate in advance”. We write that DNs “predict” certain features of behavior. We use this term because (1) these DNs correlate with certain features of behavior, and (2) changes in DN activity precede changes in behavior.

The notion that neurons can “predict” behavior is not original to our study. Whenever neuroscientists summarize the relationship between neural activity and behavior by fitting a mathematical model (which may be as simple as a linear regression), the fitted model can be said to represent a “prediction” of behavior. These models are evaluated by comparing their predictions with measured behaviors. A good model is predictive, but it also implies that the underlying neural signal is also predictive (Levenstein et al., 2023 Journal of Neuroscience 43: 1074-1088; DOI: 10.1523/JNEUROSCI.1179-22.2022). Here, prediction simply means correlation, without necessarily implying causation. We also use “prediction” to imply correlation.

We do not think the term “prediction” implies determinism. Meteorologists are said to predict the weather, but it is understood that their predictions are probabilistic, not deterministic. Certainly, we would not claim that there is a deterministic relationship between DN activity and behavior. Figure 2D shows that neither DN type can explain all the variance in the fly’s rotational or sideways velocity. At the same time, both DNs have significant predictive power.

We might equally say that these DNs “encode” behavior. We have chosen to use the word “predict” rather than “encode” because we do not think it is necessary to use the framework of symbolic communication in connection with these DNs.

We agree with the Reviewer that it is helpful to test whether any neuron that “predicts” a behavior might also “cause” this behavior. In Figure 8, we show that directly perturbing these DNs can indeed alter locomotor behavior, which suggests a causal role. Connectome analyses also suggest a causal role for these DNs in locomotor behavior (Figure 1B, see especially also Cheong et al., 2024).

At the same time, it is clear from our results that these DNs are not “command neurons” for turning: they do not deterministically cause turning. Therefore, to avoid misunderstanding, we have generally been careful to summarize the results of our perturbation experiments by avoiding the statement that “this DN causes this behavior”. Rather, we have generally tried to say that “this DN influences this behavior”, or “this DN promotes this behavior”.

(2) There is some concern about how the linear filter models were developed and then used to predict the relationship between firing rate and steering behavior: how exactly were the build and test data separated to avoid re-extracting the input? It reads like a self-fulfilling prophecy/tautology.

We used conventional cross-validation for model fitting and evaluation. We apologize that this was not made explicit in our original submission; this was due to an oversight on our part. To be clear: linear filters were computed using the data from the first 20% of a given experiment. We then convolved each cell’s firing rate estimate with the computed Neuron→Behavior filter (the “forward filter”) using the data from the final 80% of the experiment, in order to generate behavioral predictions. Thus, when a model has high variance explained, this is not attributable to overfitting: rather, it quantifies the *bona fide* predictive power of the model. We have added this information to the Methods (under “Data analysis - Linear filter analysis”).

(3) Type-O right above Figure 2 [now Figure 1E]: I assume spike rate fluctuations in DNa02 precede DNa01?

Fixed. Thank you for reading the manuscript carefully.

(4) The description of the other manuscripts about neural control of the steering as "follow-up" papers is a bit diminishing. They were likely independent works on a similar theme that happened afterwards, rather than deliberate extensions of this paper, so "subsequent" might be a more accurate description.

We apologize, as we did not intend this to be diminishing. Given this request, we have revised “follow-up” to “subsequent”.

(5) The idea that DNa02 is high-gain because it is more directly connected to motor neurons is a hypothesis and this should be made clear. We really don't know the functional consequences of the directness of a path or the number of synapses, and which circuits you compare to would change this. DNa02 may be a higher gain than DNa01, but what about relative to the other DNs that enter pre-motor regions? How do you handle a few synapses and several neurons in a common class? All of these connectivity-based deductions await functional tests - like yours! I think it is better to make this clear so readers don't assume a higher level of certainty than we have.

The Reviewer asks how we handled few-synapse connections, and how we combined neurons in the same class. We apologize for not making this explicit in our original submission. We have now added this information to the Methods. Briefly, to select cell types for inclusion in Figures 7C, we identified all individual cells postsynaptic to PFL3 and presynaptic to DNa02, discarding any unitary connections with <5 synapses. We then grouped unitary connections by cell type, and then summed all synapse numbers within each connection group (e.g., summing all synapses in all PFL3→LAL126 connections). We then discarded connection groups having <200 synapses or <1% of a cell type’s pre- or postsynaptic total. Reported connection weights are per hemisphere, i.e. half of the total within each connection group. For Figure 7F we did the same, but now discarding connection groups having <70 synapses or <0.4% of a cell type’s pre- or postsynaptic total. In Figure S9, we used the same procedures for analyzing connections onto DNa01.

We agree that it is tricky to infer function from connectome data, and this applies to motor neuron connectivity. We bring up DN connectivity onto motor neurons in two places. First, in the Results, we note that “steering filters (i.e., rotational and sideways velocity filters) were larger for DNa02 (Fig. 2A,B). This means that an impulse change in firing rate predicts a larger change in steering for this neuron. In other words, this result suggests that DNa02 operates with higher gain. This *may be related* to the fact that DNa02 makes more direct output synapses onto motor neurons (Fig. 1B) [emphasis added].” We feel this is a relatively conservative statement.

Subsequently, in the Discussion, we ask, “why does DNa02 precede larger and more rapid steering events? This may be due to the fact that DNa02 receives stronger and more direct input from key steering circuits in the brain (Fig. S9). It *may also relate* to the fact that DNa02 has more direct connections onto motor neurons (Fig. 1B) [emphasis added].” Again, we feel this is a relatively conservative statement.

To be sure, none of the motor neurons postsynaptic to DNa02 actually receive most of their synaptic input from DNa02 (or indeed any DN), and this is typical of motor neurons controlling leg muscles. Rather, leg motor neurons tend to get most of their input from interneurons rather than motor neurons (Cheong et al. 2024). Available data suggests that the walking rhythm originates with intrinsic VNC central pattern generators, and the DNs that influence walking do so, in large part, by acting on VNC interneurons. These points have been detailed in recent connectome analyses (see especially Cheong et al. 2024).

We are reluctant to broaden the scope of our connectome analyses to include other DNs for comparison, because we think these analyses are most appropriate to full-central-nervous-system-(CNS)-connectomes (brain and VNC together), which are currently under construction. Without a full-CNS-connectome, many of the DN axons in the VNC cannot be identified. In the future, we expect that full-CNS-connectomes will allow a systematic comparison of the input and output connectivity of all DN types, and probably also the tentative identification of new steering DNs. Those future analyses should generate new hypotheses about the specializations of DNa02, DNa01, and other DNs. Our study aims to help lay a conceptual foundation for that future work.

(6) Given the emphasis on the DNa02 to Motor Neuron connectivity shown (Figure 1B) and multiple text mentions, could you include more analyses of which motor neurons are downstream and how these might be expected to affect leg movements? I would like to see the synapse numbers (Figure 1B) as well as the fraction of total output synapses. These additions would help understand the evidence for the "see-saw" model.

We agree this is interesting. In follow-up work from our lab (Yang et al., 2023), we describe the detailed VNC connectivity linking DNa02 to motor neurons. We refer the Reviewer specifically to Figure 7 of that study (https://www.cell.com/cell/fulltext/S0092-8674(24)00962-0).

We regret that the see-saw model was perhaps not clear in our original submission. Briefly, this model proposes that an increase in excitatory synaptic input to one DN (and/or a disinhibition of that DN) is often accompanied by an increase in inhibitory synaptic input to the contralateral DN. This model is motivated by connectome data on the brain inputs to DNa02 (Figure 7), along with our observation that excitation of one DN is often accompanied by inhibition of the contralateral DN (Figure 5). We have now added text to the Results in several places in order to clarify these points.

This model specifically pertains to the brain inputs to DNs, comparing the downstream targets of these DNs in the VNC would not be a test of this hypothesis. The Reviewer may be asking to see whether there is any connectivity in the brain from one DN to its contralateral partner. We do not find connections of this sort, aside from multisynaptic connections that rely on very weak links (~10 synapses per connection). Figure 7 depicts a much stronger basis for this hypothesis, involving feedforward see-saw connections from PFL3 and MBON32.

(7) The conclusions from the data in Figure 8 could be explained more clearly. These seem like small effect sizes on subtle differences in leg movements - maybe like what was seen in granular control by Moonwalker's circuits? Measuring joint angles or step parameters might help clarify, but a summary description would help the reader.

We agree that these results were not explained very well in our original submission.

In our revised manuscript, we have added a new paragraph to the end of this Results section providing some summary and interpretation:

“All these effects are weak, and so they should be interpreted with caution. However, they suggest that both DNs can lengthen the stance phase of the ipsilateral back leg, which would promote ipsiversive turning. These results are also compatible with a scenario where both DNs decrease the step length in the ipsilateral legs, which would also promote ipsiversive turning. Step frequency does not normally change asymmetrically during turning, so the observed decrease in step frequency during optogenetic inhibition may just be a by-product of increasing step length when these DNs are inhibited.”

Moreover, in the Discussion, we have also added a new paragraph that synthesizes these results with other results in our study, while also noting the limitations of our study:

“Our study does not fully answer the question of how these DNs affect leg kinematics, because we were not able to simultaneously measure DN activity and leg movement. However, our optogenetic experiments suggest that both DNs can lengthen the stance phase of the ipsilateral back leg (Fig. 8G), and/or decrease the step length in the ipsilateral legs (Fig. 8H), either of which would promote ipsiversive turning. If these DNs have similar qualitative effects on leg kinematics, then why does DNa02 precede larger and more rapid steering events? This may be due to the fact that DNa02 receives stronger and more direct input from key steering circuits in the brain (Fig. S9). It may also relate to the fact that DNa02 has more direct connections onto motor neurons (Fig. 1B).”

In Figure 8D-H, we measure step parameters in freely walking flies during acute optogenetic inhibition of DNa01 and DNa02. In experiments measuring neural activity in flies walking on a spherical treadmill, we did not have a way to measure step parameters. Subsequently, this methodology was developed by Yang et al. (2023) and results for DNa02 are described in that study.

**Reviewer #3 (Recommendations for the authors):**
Minor Points:(1) If space allows, actual membrane potential should be mentioned when raw recordings are shown (for example Figure 1D).

We have now added absolute membrane potential information to Figure 1d.

(2) Typo in the sentence "To address this issue directly, we looked closely at the timing of each cell's recruitment in our dual recordings, and found that spike rate fluctuations in DNa02 typically preceded the spike rate fluctuations in DNa02 (Fig. 2A)." The final word should be "DNa01".

Fixed. Thank you for reading the manuscript carefully.

(3) Figure 2A - although there aren't direct connections between a01 and a02 in the connectome, the authors never rule out functional connectivity between these two. Given a02 precedes a01, shouldn't this be addressed?

In the full brain FAFB data set, there are two disynaptic connections from DNa02 onto the ipsilateral copy of DNa01. One connection is via CB0556 (which is GABAergic), and the other is via LAL018 (which is cholinergic). The relevant DNa02 output connections are very weak: each DNa02→CB0556 connection consists of 11 synapses, whereas each DNa02→LAL018 connection consists of 10 synapses (on average). Conversely, each CB0556→DNa01 connection consists of 29 synapses, whereas each LAL018→DNa01 connection consists of 64 synapses. In short, LAL018 is a nontrivial source of excitatory input to DNa01, but DNa02 is not positioned to exert much influence over LAL018, and the two disynaptic connections from DNa02 onto DNa01 also have the opposite sign. Thus, it seems unlikely that DNa02 is a major driver of DNa01 activity. At the same time, it is difficult to completely exclude this possibility, because we do not understand the logic of the very complicated premotor inputs to these DNs in the brain. Thus, we are hesitant to make a strong statement on this point.